

# Review of the global models used within the Chemistry-Climate Model Initiative (CCMI)

Olaf Morgenstern[1], Michaela I. Hegglin[2], Eugene Rozanov[18,5], Fiona M. O'Connor[14], N. Luke Abraham[17,20], Hideharu Akiyoshi[8], Alexander T. Archibald[17,20], Slimane Bekki[21], Neal Butchart[14], Martyn P. Chipperfield[16], Makoto Deushi[15], Sandip S. Dhomse[16], Rolando R. Garcia[7], Steven C. Hardiman[14], Larry W. Horowitz[13], Patrick Jöckel[10], Beatrice Josse[9], Douglas Kinnison[7], Meiyun Lin[13], Eva Mancini[3], Michael E. Manyin[12], Marion Marchand[21], Virginie Marécal[9], Martine Michou[9], Luke D. Oman[12], Gianni Pitari[3], David A. Plummer[4], Laura E. Revell[5,6], David Saint-Martin[9], Robyn Schofield[11], Andrea Stenke[5], Kane Stone[11,*], Kengo Sudo[19], Taichu Y. Tanaka[15], Simone Tilmes[7], Yousuke Yamashita[8,#], Kohei Yoshida[15], and Guang Zeng[1]

[1]National Institute of Water and Atmospheric Research (NIWA), Wellington, New Zealand
[2]Department of Meteorology, University of Reading, UK
[3]Department of Physical and Chemical Sciences, Universitá dell'Aquila, Italy
[4]Environment and Climate Change Canada, Montréal, Canada
[5]Institute for Atmospheric and Climate Science, ETH Zürich (ETHZ), Switzerland
[6]Bodeker Scientific, New Zealand
[7]National Center for Atmospheric Research (NCAR), Boulder, Colorado, USA
[8]National Institute of Environmental Studies (NIES), Tsukuba, Japan
[9]CNRM UMR 3589, Météo-France/CNRS, Toulouse, France
[10]Institut für Physik der Atmosphäre, Deutsches Zentrum für Luft- und Raumfahrt (DLR), Oberpfaffenhofen, Germany
[11]School of Earth Sciences, University of Melbourne, Victoria, Australia
[12]National Aeronautics and Space Administration Goddard Space Flight Center (NASA GSFC), Greenbelt, Maryland, USA
[13]National Atmospheric and Ocean Administration Geophysical Fluid Dynamics Laboratory (NOAA GFDL) and Princeton University Program in Atmospheric and Oceanic Sciences, Princeton, New Jersey, USA
[14]Met Office Hadley Centre (MOHC), Exeter, UK
[15]Meteorological Research Institute (MRI), Tsukuba, Japan
[16]School of Earth and Environment, University of Leeds, UK
[17]Department of Chemistry, University of Cambridge, UK
[18]Physikalisch-Meteorologisches Observatorium Davos - World Radiation Center (PMOD/WRC), Davos, Switzerland
[19]Graduate School of Environmental Studies, Nagoya University, Japan
[20]National Centre for Atmospheric Science (NCAS), UK
[21]LATMOS, Institut Pierre Simon Laplace (IPSL), Paris, France
*now at Massachusetts Institute of Technology (MIT), Boston, Massachusetts, USA
#now at Japan Agency for Marine-Earth Science and Technology (JAMSTEC), Yokohama, Japan

*Correspondence to:* Olaf Morgenstern (olaf.morgenstern@niwa.co.nz)

**Abstract.**

We present an overview of state-of-the-art chemistry-climate and -transport models that are used within the Chemistry-Climate Model Initiative (CCMI). CCMI aims to conduct a detailed evaluation of participating models using process-oriented diagnostics derived from observations in order to gain confidence in the models' projections of the stratospheric ozone layer, air quality, where applicable global climate change, and the interactions between them. Interpretation of these diagnostics



requires detailed knowledge of the radiative, chemical, dynamical, and physical processes incorporated in the models. Also an understanding of the degree to which CCMI recommendations for simulations have been followed is necessary to understand model response to anthropogenic and natural forcing and also to explain inter-model differences. This becomes even more important given the ongoing development and the ever-growing complexity of these models. This paper also provides an

overview of the available CCMI simulations with the aim to inform CCMI data users.

## 1   Introduction

Climate models have been evolving considerably in recent decades. From relatively simple beginnings, ever more components have been added, until presently "Earth System Models" (ESMs) define the state of the field. Such models simultaneously serve a variety of purposes including simulating air quality, stratospheric ozone, and global climate. These applications are

strongly coupled; e.g., various air pollutants are climate active, stratospheric and tropospheric ozone are coupled in a variety of ways, and climate change affects atmospheric composition and vice versa. Previous-generation models were generally not that multi-faceted and were usually constructed for just one of these purposes. Also increasingly biogeochemical feedbacks are considered, e.g. in the form of organic aerosol precursors emitted from land and ocean surfaces. However, the additional complexity characterizing ESMs makes these simulations more difficult to evaluate because previously ignored feedbacks now

need to be considered.

     The purpose of this paper is to document the internal make-up of 20 chemistry-climate models (CCMs) and other models participating in the Chemistry-Climate Model Initiative (CCMI, Eyring et al., 2013), a combined activity of the International Global Atmospheric Chemistry (IGAC) and Stratosphere-troposphere Processes and their Role in Climate (SPARC) projects. CCMs are a major stepping stone on the way towards ESMs, combining physical climate models with an explicit representa-

tion of atmospheric chemistry. CCMI continues the legacy of previous chemistry-climate model intercomparisons, particularly the Chemistry-Climate Model Validation (CCMVal, SPARC, 2010) and the Atmospheric Chemistry and Climate Model Intercomparison Project (ACCMIP, Lamarque et al., 2013) activities. These precursor activities had more limited aims than CCMI: In the case of CCMVal, the main aim was to inform the World Meteorological Organization's Scientific Assessments of Ozone Depletion (WMO, 2007, 2011, 2015) with state-of-the-science information about stratospheric ozone and its past

and projected future evolution. The main outcome was SPARC (2010), a comprehensive assessment of the performance of stratospheric chemistry-climate models. In the case of ACCMIP, the aim was to inform the 5[th] Assessment Report of the Intergovernmental Panel on Climate Change (IPCC, 2013) about the roles of "near-term climate forcers", notably tropospheric ozone and aerosols, in historical and future climate change. CCMI builds on CCMVal and ACCMIP by encouraging the participation of coupled atmosphere-ocean stratosphere-troposphere models that represent the various ways in which stratospheric

and tropospheric composition are coupled to each other and to the physical climate more consistently than in their predecessor models.

     For CCMVal, Morgenstern et al. (2010) describe salient model features of CCMVal-2 models, mostly in tabular forms. The paper builds on numerous publications that describe individual models, or aspects thereof. Here we present an update to the



tables in Morgenstern et al. (2010), focussing in the accompanying text on the changes that have occurred in the participating models since CCMVal-2. (In most cases, older versions of the CCMI models had participated in CCMVal-2, see below.) This paper is meant to support other publications evaluating the CCMI simulations by providing an overview of the make-up of CCMI models as well as a comprehensive literature list for further reading.

## 2 Participating models

There are 20 models participating in CCMI (cf. tables 1 and 2). With three exceptions (CHASER, TOMCAT, MOCAGE), each participating model had a predecessor model in CCMVal-2; hence, the focus here will be on developments since CCMVal-2. Corresponding to the much broader scope of CCMI, relative to CCMVal-2, salient developments in these models include whole-atmosphere chemistry (almost all CCMVal-2 models have been further developed to include tropospheric chemistry), coupling (now several of the CCMI models include an interactive ocean), increased resolution for several of them, and progress in various areas of model physics.

## 3 Model details

In the following, we comment on noteworthy developments relative to the CCMVal-2 models (SPARC, 2010). Apart from some very high-level information (such as model names and contact information), all tabulated information about the models is in the supplementary material.

### 3.1 General model make-up

For CCMVal-2, all but one model (CMAM) were atmosphere-only. In the CCMI ensemble, there are now 9 models which at least for some simulations are coupled interactively to ocean and sea ice models (CESM1 CAM4-chem, CESM1 WACCM, CHASER, EMAC, GFDL-CM3, HadGEM3-ES, LMDz-REPROBUS-CM6, MRI-ESM, NIWA-UKCA; see supplement, table S1). This means that in these models, e.g. the impact of ozone depletion on surface climate is represented consistently. Like in CCMVal-2, some family relationships are apparent between different models. For example, ACCESS-CCM, NIWA-UKCA, UMUKCA-UCAM, and HadGEM3-ES use similar atmosphere, ocean and sea ice components (ACCESS-CCM and UMUKCA-UCAM are atmosphere-only). GFDL-AM3 is the atmosphere-only equivalent of GFDL-CM3. EMAC and SOCOL are both based on different versions of the ECHAM5 climate model. LMDz-REPROBUS-CM6 can be coupled to a similar version of the NEMO ocean model as NIWA-UKCA and HadGEM3-ES; however at the time of writing only the atmosphere-only CM5 version has been used for CCMI simulations. CCSRNIES uses a similar version of the MIROC atmosphere model as CHASER.

In CCMVal-2 one model (AMTRAC) used a novel grid which was neither latitude-longitude nor spectral, namely the cubed-sphere grid. While several modelling centres presently are working on new-generation models based on this or similar novel grids, in the CCMI ensemble the GFDL successors to AMTRAC continue to use this grid; GEOSCCM has adopted this grid



(table 3). It is anticipated that more models will adopt novel grids in future model intercomparisons, to improve scalability and computing efficiency.

## 3.2 Model resolution

The horizontal resolution of most models is unchanged versus CCMVal-2 (table 3). ULAQ CCM, HadGEM3-ES, MRI ESM, CMAM, CNRM-CM5-3 (for chemistry), SOCOL, and LMDz-REPROBUS-CM6 have improved their horizontal resolution; now resolution ranges between roughly 5° to less than 2°. In several cases, models have improved their vertical resolution, particularly CNRM-CM5-3, LMDZ-REPROBUS, MRI ESM, ULAQ CCM, and HadGEM3-ES (table 3). Vertical ranges are essentially unchanged versus the models' CCMVal-2 counterparts with the uppermost model levels ranging from 35 km (for MOCAGE) to 140 km (for CESM1 WACCM). All but two models (CESM1 CAM4-chem and MOCAGE) completely cover the stratosphere. The vertical resolution varies with altitude range and model (cf. supplement, table S2).

## 3.3 Quasi-biennial oscillation

Essentially, the same models that used nudging for CCMVal-2 continue to use nudging to impose a Quasi-biennial osciallation (QBO) in their models. The nudging is performed in a more sophisticated way, with SOCOL, CCSRNIES, and EMAC now using smooth transitions at the edges of the nudged region (supplement, table S3). Other models do not impose a QBO (except for the specified-dynamics simulations), which means the QBO may not require explicit forcing to occur in the models, or it may be absent.

## 3.4 Volcanic effects

There has been considerable progress regarding the physical consistency of volcanic effects in CCMs. Whereas in CCMVal-2, surface area densities and aerosol-induced heating rates were prescribed, now 10 of the CCMI models treat radiative effects on-line, i.e. calculate or assume an aerosol size distribution for volcanic aerosol in the stratosphere and derive radiative heating rates from these (supplement, table S4). A few of these models do not couple this volcanic aerosol to radiation or prescribe heating rates associated with the presence of this aerosol (GFDL-AM3/CM3, CNRM-CM5-3, LMDz-REPROBUS, TOMCAT, UM-SLIMCAT). Surface area density used in heterogeneous chemistry calculations, with only two exceptions (GFDL-AM3/CM3, UMUKCA-UCAM), follows the CCMI recommendation (section 5).

## 3.5 Advection

In CCMVal-2, there were some models that used different transport schemes for hydrological versus chemical tracers (Morgenstern et al., 2010). For CCMI all chemistry-climate models (which transport both types of tracers) employ the same transport scheme for both (supplement, tables S5, S26). This makes the advection of all tracers physically self-consistent. The Met Office's Unified Model (MetUM) family (ACCESS CCM, HadGEM3-ES, NIWA-UKCA, UMSLIMCAT, UMUKCA-UCAM)





still uses different settings for hydrological and chemical tracers versus physical tracers (momentum, heat). All other models treat these physical tracers consistently with other tracers.

### 3.6  Timestepping and calendars

Atmosphere models use a variety of timesteps for the dynamical core, physical processes, radiation, transport, chemistry, and
the coupling of chemistry and dynamics for different reasons. Generally the choice of timestep is the result of a compromise between the computational cost associated with short timesteps and the reduced accuracy associated with larger timesteps. There is a considerable amount of diversity in the CCMI ensemble regarding the choices of timesteps for different processes (supplement, table S6). In addition, most models now use the Gregorian or the 365-day calendars (whereas in CCMVal-2, for reasons of easier handling of averages and climatologies, often a 360-day calendar was used). Only the MetUM based
models (ACCESS CCM, HadGEM3-ES, NIWA-UKCA, UMSLIMCAT, UMUKCA-UCAM) and ULAQ still use the 360-day calendar.

### 3.7  Physical parameterizations

References for the descriptions of the models' physical parameterizations such as turbulent vertical fluxes and dry convection, moist convection, cloud microphysics, aerosol microphysics, and cloud cover can be found in the supplement, tables S7 and
S8. Several models have renewed their physics parameterizations since CCMVal-2, namely ACCESS CCM, NIWA-UKCA, CNRM-CM5-3, GFDL-CM3/AM3, HadGEM3-ES, LMDz-REPROBUS, MRI-ESM, SOCOL, and UMUKCA-UCAM.

### 3.8  Tropospheric aerosols

Most models now included an interactive aerosol scheme, except for CCSRNIES, EMAC, CNRM-CM5-3, LMDz-REPROBUS, and SOCOL. CMAM uses prescribed sulfate aerosol surface area densities in the troposphere for heterogeneous chemistry cal-
culations. In CCMVal-2, most models did not have any representation of tropospheric aerosols. In CCMI, various approaches of differing complexity are used (supplement, table S8).

### 3.9  Stratospheric chemistry

Stratospheric gas-phase chemistry is well-enough understood and sufficiently simple so that it can be treated mostly explicitly, by adopting all relevant reactions for which rate coefficients have been published. Most models follow the Sander et al.
(2011b) recommended rates. There is some diversity as to which halogen source gases are considered (supplement, table S9). Several models represented here also participated in the re-assessment of lifetimes of long-lived species (SPARC, 2013) and in this context have expanded their range of halogen source gases. Most Unified Model based participants (ACCESS CCM, HadGEM3-ES, NIWA-UKCA, UMSLIMCAT, not UMUKCA-UCAM) continue to lump all chlorine source gases into only two representatives (CFC-11, CFC-12). SOCOL and the CESM model family, at 14 and 12 species including Halon-1211,
respectively, have the largest number of chlorine source gases. For bromine, the recommendation was to include the short-lived



constituents di-bromomethane ($CH_2Br_2$) and bromoform ($CHBr_3$); about half of the models follow this recommendation. All models represent $CH_3Br$ (the most abundant bromine source gas); several also include Halon-1211, Halon-1301, and/or Halon-2402. EMAC also has a representation of a sea salt aerosol source of gas-phase halogen which may be of importance to the tropospheric oxidizing capacity (Allen et al., 2007), and a larger range of very short-lived organic bromine compounds

which likely influence tropospheric and stratospheric ozone chemistry. There do not appear to be fundamental differences w.r.t. how stratospheric chemistry was treated for CCMVal-2. Morgenstern et al. (2010) provide a more exhaustive discussion of stratospheric chemistry in CCMVal-2 models that is still generally relevant.

### 3.10 Tropospheric chemistry

In contrast to CCMVal-2 (which did not focus on tropospheric chemistry), a majority of CCMI models now explicitly represent

tropospheric ozone chemistry (supplement, table S9). Six models do not represent any non-methane hydrocarbon (NMHC) chemistry (CCSRNIES MIROC3.2, CMAM, CNRM-CM5-3, LMDz-REPROBUS, TOMCAT, UMSLIMCAT). In LMDz-REPROBUS, climatological zonally invariant constituent fields are assumed below 400 hPa. Unlike stratospheric chemistry, tropospheric chemistry is too complex to incorporate comprehensively in a chemistry-climate model. The need to include an affordable yet skilled tropospheric chemistry scheme drives some diversity in the chemistry schemes and correspondingly the

represented NMHC source gases. Several schemes use lumping, whereby emissions of a non-represented NMHC source gas are implemented as emissions of a represented one. Sometimes this species is denoted as "a lumped species" or "OTHC" (other carbon; supplement, table S25). SOCOL has the simplest organic chemistry scheme in the ensemble (the only organic NMHC source gas, disregarding HCHO, is isoprene, $C_5H_8$). By contrast, the chemistry-transport model MOCAGE and several CCMs represent 10 or more NMHC source gases.

For methane ($CH_4$) the recommendation is to use a single prescribed time-evolving volume mixing ratio (as defined by the Representative Concentration Pathway (RCP 6.0), Meinshausen et al., 2011) as the global lower boundary condition. This is followed by almost all models. CHASER has an interactive methane scheme, and the EMAC and ULAQ models prescribe $CH_4$ at the surface under consideration of a hemispheric asymmetry (i.e. there is about 5% less methane in the Southern than in the Northern Hemisphere). EMAC also prescribes a seasonal cycle for $CH_4$.

### 3.11 Stratospheric and tropospheric heterogeneous chemistry

Heterogeneous chemistry (i.e. reactions that require a solid or liquid surface as a catalyst) is crucial to several aspects of atmospheric chemistry, notably the ozone hole and the tropospheric nitrogen cycle. Most of the reactions in the supplement, table S10, are chlorine and/or bromine activation reactions, e.g. they turn chlorine from its unreactive forms (HCl, $ClONO_2$) into reactive forms (that photolyze readily in sunlight). The implementation of heterogeneous chemistry is subject to consid-

erable inter-model differences regarding represented reactions and their associated heterogeneous surface types. Seven models (CCSRNIES MIROC3.2, CESM1 CAM4-chem, CESM1 WACCM, CHASER, CMAM, GFDL CM3/AM3, and MOCAGE) explicitly consider supercooled ternary solutions (STS; mixtures of $HNO_3$, $H_2SO_4$, and $H_2O$) which impact stratospheric chemistry through swelling of droplets and associated denitrification and heterogeneous chemistry. In the troposphere, nitric



acid formation (and subsequent nitrogen removal) partly occurs on/in cloud droplets and on aerosol surfaces. Models vary greatly in how this process is implemented. Also $SO_2$ oxidation to form $SO_3$ (which then further reacts to form sulfate aerosol) partly occurs in the aqueous cloud phase. Most models include this heterogeneous reaction (supplement, table S28).

### 3.12 Ocean surface forcing

For the atmosphere-only reference (REF-C1 and REF-C1SD) and sensitivity (SEN-C1) simulations (section 4), ocean surface forcing (sea surface temperatures, sea ice) need to be imposed (Eyring et al., 2013). Most modelling groups used the Hadley Centre Ice and Sea Surface Temperature (HadISST) dataset (Rayner et al., 2003), as recommended, but also other datasets were used (supplement, table S11). For the atmosphere-ocean coupled reference (REF-C2) and sensitivity (SEN-C2) simulations (section 4), those models that do not couple to an interactive ocean/sea ice module require climate model fields to be imposed.

A substantial variety of different climate model datasets were used for this purpose. In the ULAQ CCM simulations, an ocean surface dataset was used that was derived from a climate model simulation, with mean biases relative to HadISST removed.

### 3.13 Horizontal diffusion

Numerical diffusion is an unavoidable aspect of numerical climate models, linked to the discrete nature of grids used for transport processes. Transport schemes are generally designed to minimize numerical diffusion. However, in addition, several

models require explicit diffusion for stability (supplement, table S12). In CESM1 CAM4-chem and CESM1 WACCM, hyperdiffusion is applied to the smallest scales, and through Fourier transformation and filtering the effective resolution is kept the same at all latitudes. Several models (ACCESS CCM, NIWA-UKCA, UMUKCA-UCAM, GEOSCCM, HadGEM3-ES, LMDz-REPROBUS, MOCAGE) do not contain explicit diffusion in most of their domains. "Sponges" are generally used to prevent reflection of planetary or Rossby waves off the model top, except for CMAM, the MRI ESM, and HadGEM3-ES.

The MetUM family of models also requires diffusion over the poles (ACCESS CCM, NIWA-UKCA, UMUKCA-UCAM, HadGEM3-ES). The need for polar filtering should disappear with the future adoption of "novel" grids that no longer have any singularities at the poles.

### 3.14 Orographic and non-orographic gravity wave drag

Gravity waves are the result of vertical displacements of air in the presence of stratification, which can be due e. g. to mountains,

frontal systems, or tropospheric convective activity. They can either be dissipated if they encounter critical levels (at which the phase speed equals the background winds), or they continue to propagate upwards and increase in amplitude, in accordance with the decreasing air density. Eventually they can break, leading to deceleration of the mean flow. This process contributes to the driving of the stratospheric Brewer-Dobson Circulation, but also affects the temperature structure of the middle atmosphere. Their horizontal scale is mostly below the grid scale, meaning that this process needs to be parameterized. Gravity waves are

also poorly observed, contributing to a substantial diversity of approaches to representing this process (supplement, table S13).



The paucity of observations leads to gravity wave drag being often used to tune better known model diagnostics such as stratospheric temperatures or age of air.

Gravity wave drag (GWD) is usually divided into two components for modelling: Orographic and non-orographic drag. The representation of orographic drag is based on the interaction of flow with topography, a relatively well-known process.
Non-orographic gravity waves by contrast are geographically poorly constrained, so often relatively simple approaches, not taking into account any tropospheric meteorology, are used. However, in contrast to CCMVal-2, several models now link non-orographic drag to tropospheric processes such as convection (CNRM-CM5-3, CESM1 WACCM). This means that in these models, possible changes in the GWD sources associated with climate change are represented.

### 3.15  Cloud microphysics

Clouds remain a very substantial source of uncertainty and inter-model differences e. g. regarding climate sensitivity. Small-scale variability, non-equilibrium processes, cloud-aerosol interactions, and other processes all contribute to this. The CCMI model ensemble is characterized by some considerable diversity in approaches to cloud microphysics (supplement, table S14), and most models have implemented changes in the way clouds are represented, relative to CCMVal-2 (where clouds never were a particular focus).

### 3.16  Land surface, soil, and the planetary boundary layer

Land surface properties, such as vegetation and soil type, but also soil moisture, snow, and groundwater have significant climate effects (e. g. influence the severity of droughts and floods), mediated through surface albedo and evapotranspiration (IPCC, 2013). Hence their representation in climate models are essential especially for regional climate simulations. CCMI models, like IPCC-type climate models, generally have land surface schemes. Like for cloud microphysics, there is some considerable diversity of approaches in treating these three aspects of the models (supplement, table S15).

### 3.17  Polar stratospheric clouds (PSCs)

Like for CCMVal-2, the models divide into two groups: Those that assume thermodynamical equilibrium for PSCs, and others (CESM1, EMAC, GEOSCCM, ULAQ) that account for deviations from thermodynamic equilibrium (supplement, table S16). With the exception of CMAM, all models account for PSC sedimentation (which leads to denitrification) but assumptions around this vary considerably. Several models impose fixed sedimentation velocities for the different PSC types; in others, these velocities are a function of particle size. In most models, the approach to handling PSCs appears to be unchanged versus SPARC (2010).





### 3.18 Shortwave radiation

In most cases, models are using the same basic schemes as documented in SPARC (2010, supplement, table S17,). However, ULAQ, MRI ESM1r1, the 79-level version of LMDz-REPROBUS, the GFDL models, EMAC, CNRM-CM5-3, CCSRNIES MIROC3.2, and SOCOL have increased their spectral resolution versus their CCMVal-2 predecessors.

### 3.19 Longwave radiation

Longwave radiation is treated largely in the same way as documented in (SPARC, 2010). However, again a few models (CNRM-CM5-3, MRI-ESM1r1, SOCOL, ULAQ) have increased their spectral resolution versus their CCMVal-2 predecessors (supplement, table S18).

### 3.20 Photolysis

In CCMVal-2, half the models used tabulated photolysis rates and interpolation to calculate photolysis rates (Morgenstern et al., 2010). This approach is problematic in the troposphere because of complicating effects of clouds, aerosols, surface albedo, and other factors that are not considered in the pre-calculated tables. It is however computationally more efficient. With the new focus, relative to CCMVal-2, on tropospheric chemistry, all models that have explicit tropospheric chemistry also take explicit account of the presence of clouds (supplement, table S19). The MetUM family (ACCESS CCM, HadGEM3-ES, NIWA-UKCA, UMUKCA-UCAM) has adopted the Fast-JX online formulation of photolysis in the domain below 60 km, and a group of other models continue to use look-up tables but apply corrections accounting for the presence of clouds (CESM1 CAM4-chem, CESM1 WACCM, CMAM, MOCAGE, MRI-ESM1r1, SOCOL). The GFDL-AM3 model applies the Fast-JX photolysis scheme. Also in many cases photolysis cross sections have been updated, relative to CCMVal-2.

### 3.21 Tropospheric aerosol

The additional focus, relative to CCMVal-2, on tropospheric climate-composition linkages has led to most models including an explicit treatment of tropospheric aerosol. Most schemes are "bulk", i.e. only total mass of an aerosol type is predicted (supplement, table S20). In bulk schemes, derived quantities such as particle number require assumptions about particle sizes to be made. The ULAQ CCM and MRI-ESM1r1 use sectional approaches which represent aerosols of different size classes in discrete bins, thus avoiding a-priori assumptions on particle size. The ULAQ CCM represents nitrate aerosol in a modal way, i.e. the aerosol size distribution is assumed to be described by one or more log-normal distributions. Modal schemes are computationally more efficient than sectional schemes while also predicting both aerosol size and number. Several models (CNRM-CM5-3, EMAC, SOCOL) use off-line representations of aerosol.

### 3.22 Ocean coupling

Eight of the CCMI models are coupled, at least for some simulations, to an interactive ocean module (supplement, table S21), namely CESM1 CAM4-chem, CESM1 WACCM, CHASER, EMAC, HadGEM3-ES,LMDz-REPROPUS-CM6, MRI-ESM



1r1, and NIWA-UKCA. This is a substantial increase from CCMVal-2, when only one model (CMAM) was coupled to an ocean model. Three of the models (HadGEM3-ES, LMDz-REPROBUS-CM6, NIWA-UKCA) use versions of the Nucleus of a European Model of the Ocean (NEMO). The other five use independent ocean models. The coupling between the atmosphere, ocean, and sea ice modules involves the passing of several physical fields that define the interactions between these modules,

which essentially consist of transfers of momentum, heat, moisture, and salinity. The coupling frequency also varies a lot, from daily to hourly.

### 3.23   Solar forcing

Interactions between the atmosphere and the Sun are considered in an increasing consistent manner in CCMI models (supplement, tables S29 and S30). All models consider spectrally resolved irradiance. In five models (EMAC, HadGEM3-ES,

MRI ESM1r1, SOCOL, and ULAQ CCM) photolysis and short-wave radiation are handled consistently. SOCOL and the MRI ESM1r1 also consider proton ionization by solar particles. With the exceptions of ACCESS CCM, NIWA-UKCA, and GEOSCCM, all models listed in table S29 consider solar variability.

### 4   CCMI simulations

In this section, we briefly describe the motivation and some technical details regarding the experiments conducted for CCMI.

Eyring et al. (2013) have given more details. Forcings are discussed briefly in section 5.

-   REF-C1: This experiment is analogous to the REF-B1 experiment of CCMVal-2. Using state-of-knowledge historic forcings and observed sea surface conditions, the models simulate the recent past (1960-2010). During this period, ozone-depleting substances peak roughly in the year 2000, some important types of industrial emissions in Europe and North America undergo a maximum in the 1980s, emissions from East and South Asia are growing, and there is a

considerable increase in greenhouse gases. The models are free-running.

-   REF-C1SD: This is similar to REF-C1 but the models are nudged towards reanalysis datasets and correspondingly the simulations only cover 1980-2010. Through a comparison to the REF-C1 simulations, the influence of dynamical biases on composition can be assessed. This type of experiment had not been conducted for CCMVal-2. The supplement, table S27, has details on how nudging is implemented in those models that have conducted the specified-dynamics simulations.

-   REF-C2: This experiment is a set of seamless simulations spanning the period 1960-2100, similar to the REF-B2 experiment for CCMVal-2. The experiments follow the WMO (2011) A1 scenario for ozone-depleting substances and the Representatitive Concentration Pathway (RCP) 6.0 (Meinshausen et al., 2011) for other greenhouse gases, tropospheric ozone ($O_3$) precursors, and aerosol and aerosol precursor emissions. Ocean conditions can either be taken from a separate climate model simulation, or the models can be coupled interactively to ocean and sea ice modules.

In addition to these reference simulations, a variety of sensitivity simulations have been asked for, which are variants on the reference simulations, typically with just one aspect changed.



- SEN-C1-Emis / SEN-C1SD-Emis: In these experiments the recommended emission dataset is replaced with an emission dataset of the modellers' choice, to assess the impact of alternative emissions on tropospheric composition.

- SEN-C1-fEmis /SEN-C1SD-fEmis: In these experiments, 1960 emissions are prescribed throughout, allowing the role of meteorological variability in influencing tropospheric composition to be established.

- SEN-C2-RCP: This is the same as REF-C2, but with the GHG scenario changed to either RCP 2.6, 4.5, or RCP 8.5. The simulations start in 2000.

- SEN-C2-fODS / SEN-C2-fODS2000: This is the same as REF-C2 but with ozone-depleting (halogenated) substances (ODSs) fixed at their 1960 or 2000 levels, respectively. The SEN-C2-fODS2000 simulations start in 2000.

- SEN-C2-fGHG: This is similar to REF-C2 but with GHGs fixed at their 1960 levels, and sea surface conditions prescribed as the 1955-1964 average (where these conditions are imposed).

- SEN-C2-fEmis: This is similar to REF-C2 but with surface emissions fixed to their respective 1960 levels.

- SEN-C2-GeoMIP: These simulations link CCMI with the Geoengineering Model Intercomnparison Project (GeoMIP, Tilmes, 2015a). They are designed to test the impact of proposed efforts to actively manage the Earth' radiation budget to offset the impact of increasing GHGs using sulfur injections.

- SEN-C1-SSI: This is similar to REF-C1 but using a solar forcing dataset with increased UV intensity (Krivova et al., 2006).

- SEN-C2-SolarTrend: This experiment will assess the impact of a possible reduction of solar activity akin to the Maunder Minimum of the 17th and 18th centuries.

- SEN-C2-fCH4: This experiment is identical to REF-C2 but the methane surface mixing ratio is fixed to its 1960 value (Hegglin et al., 2016).

- SEN-C2-fN2O: Same as SEN-C2-fCH4 but for $N_2O$ (Hegglin et al., 2016).

## 5   Forcings used in the reference simulations

Eyring et al. (2013) and Hegglin et al. (2016) provide full details of the forcings to be used in the above listed CCMI simulations. Here we only comment on selected aspects.

### 5.1   Greenhouse gases

Most simulations use historical and/or RCP 6.0 mixing ratios for GHGs (figure 1a). These are characterized by continuing increases of carbon dioxide ($CO_2$), which more than doubles between 1960 and 2100. However the rate of increase reduces at





**Figure 1.** Selected forcings used in the REF-C2 simulations. (a) Carbon dioxide ($CO_2$; solid), methane ($CH_4$; dashed), and nitrous oxide ($N_2O$; dash-dotted) surface mass mixing ratios, following RCP 6.0 (Meinshausen et al., 2011). (b) total chlorine (Cl; solid) and total bromine (Br) excluding the mass mixing ratios of the very short-lived species (VSLS; dashed; scenario A1 of WMO, 2011). (c) Total nitrogen oxide ($NO_x$) emissions. Solid: global. Dashed: Europe. Dotted: North America. Dash-dotted: East Asia. Dash-dot-dot-dotted: South Asia. (d) Same but for carbon monoxide (CO).

the end of this period. Nitrous oxide ($N_2O$) also increases continuously from around 290 ppbv in 1960 to over 400 ppbv in 2100. Methane ($CH_4$) increases during the 20th century, plateaus between around 2000 and 2030, and subsequently shows a renewed increase, a maximum in around 2070, and then a decrease in the last few decades of the 21st century. In comparison to the REF-B2 simulations of CCMVal-2, $CO_2$ and $N_2O$ follow similar projections, but $CH_4$ has a considerably reduced maximum

5  which also occurs later in the 21st century (SPARC, 2010, fig. 2.3).



## 5.2 Ozone-depleting substances (ODSs)

ODSs develop according to the A1 scenario of WMO (2011) (figure 1b). There are no major differences w.r.t. the scenario used by CCMVal-2 (SPARC, 2010, fig. 2.3). The A1 scenario does not take into account the revised lifetimes of ODSs as documented in SPARC (2013). Test simulations with a scenario based on these revised lifetimes indicate that there would be no significant impact on ozone (WMO, 2015); hence the recommendation for ODSs has remained unchanged. In addition to long-lived ODSs, modellers are recommended to include $CH_2Br_2$ and $CHBr_3$ as bromine source gases. Both are classified as Very Short-Lived Substances (VSLS). Surface mixing ratios for both are fixed at 1.2 pptv (giving a total of 6 pptv of bromine). Considering losses of both species in the troposphere, they are meant to essentially deliver the $\sim 5$ pptv of inorganic bromine to the stratosphere that is thought to originate as VSLS.

## 5.3 Anthropogenic tropospheric ozone and aerosol precursors

For the REF-C2 scenario, for anthropogenic emissions the recommendation is essentially to use MACCity (Granier et al., 2011) until 2000, followed by RCP 6.0 emissions. Figure 1(c,d) shows globally and regionally integrated emissions of nitrogen oxides ($NO_x$) and carbon monoxide (CO). Globally, efforts to improve air quality, introduced during the late $20^{\text{th}}$ century, cause $NO_x$ and CO emissions to peak and then decline. This is clearly seen in European and North American emissions, but East and South Asian emissions continue to increase – East Asian emissions of $NO_x$ only peak in around 2050. Notably, for $NO_x$ there is a discontinuity in 2000 caused by differences in the assumptions on ship emissions between MACCity and RCP 6.0. For REF-C1, only MACCity is used.

## 5.4 Biogenic emissions

For natural (biogenic) emissions, the CCMI recommendation is to use interactive emissions, where available. The extent to which interactive schemes are used, however, is very species- and model-dependent. In the case of soil nitrogen oxide ($NO_x$) emissions, the majority of models use prescribed emissions, with the exceptions of EMAC and GEOSCCM which both use the Yienger and Levy (1995) emissions scheme (supplement, table S22). For oceanic dimethyl sulfide (DMS) emissions, most models use interactive emissions schemes with some commonality in the choice of scheme (e.g., Wanninkhof, 1992; Chin et al., 2002), particularly within model families, although a small number of models also use prescribed emissions for oceanic DMS. For biogenic acetone (($CH_3$)$_2$CO) emissions, all but the CESM1 models either exclude ($CH_3$)$_2$CO or use prescribed emissions. CESM1 CAM4-chem and CESM1 WACCM, however, use the MEGAN2.1 interactive scheme. For ethane ($C_2H_6$), the CESM1 models also use MEGAN2.1 whereas all other models either exclude $C_2H_6$ altogether or prescribe emissions. For isoprene ($C_5H_8$) emissions, about half of the models prescribe emissions and for those that use interactive terrestrial emissions, MEGAN is predominantly the emissions scheme of choice. A small number of models include interactive oceanic $C_5H_8$ emissions.





### 5.5 Sea surface temperature and sea ice

For the REF-C1 and SEN-C1 experiments, sea surface conditions need to be prescribed. For this, as for CCMVal-2, the HadISST climatology (Rayner et al., 2003) is recommended. Variance correction for this monthly-mean climatology is recommended following the AMIP II method (http://www-pcmdi.llnl.gov/projects/amip/AMIP2EXPDSN/BCS/bcsintro.php). Between 1960 and 2010 there is some warming in the HadISST dataset over various areas of the ocean, but also widespread cooling of the Southern Ocean (figure 2). Arctic annual minimum sea ice extent reduces considerably in this period, whereas Antarctic sea ice expands slightly (figure 3). HadISST is heavily based on satellite observations which are non-existent for the early part of the record, meaning that trends derived over this period have to be viewed with caution. For the REF-C2 and SEN-C2 experiments, either an interactive ocean/sea ice sub-model is used or pre-calculated sea surface conditions derived from a variety of different climate model simulations as detailed in the previous paragraphs.

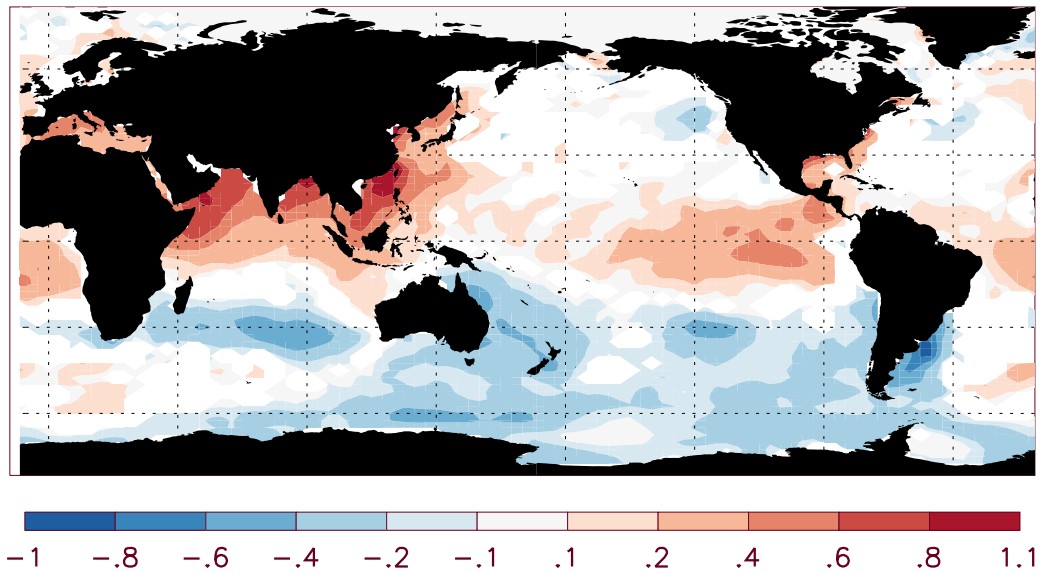

**Figure 2.** Trend in the annual-mean SSTs, in HadISST for 1960-2010 (K/century). All coloured trends are significant at the 95% confidence interval.

### 5.6 Stratospheric aerosol loading

Figure 4 shows the aerosol surface area density at 22 km as recommended by CCMI and imposed by most models (Arfeuille et al., 2013). In comparison to the data set used for CCMVal-2 (Morgenstern et al., 2010), the most significant difference is the insertion of a major volcanic injection of aerosol into the stratosphere in 1974/1975, due to the Fuego (Guatemala) eruption. This had been ignored before. Also note the increase in aerosol density during the last decade, attributed to a series of small volcanic eruptions (Vernier et al., 2011).





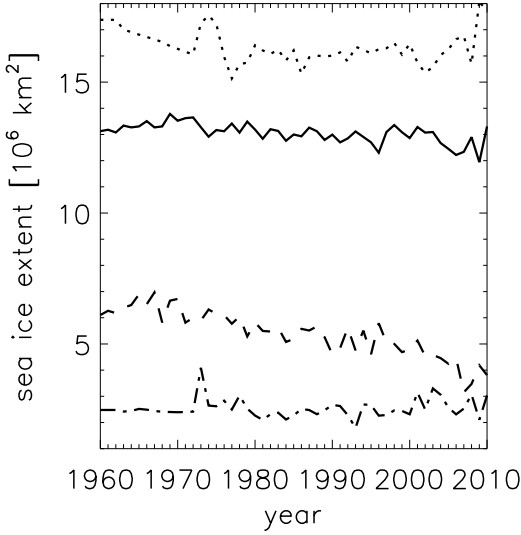

**Figure 3.** Maximum and minimum monthly-mean sea ice extent in the HadISST climatology. Dotted: Antarctic maximum. Solid: Arctic maximum. Dashed: Arctic minimum. Dot-dashed: Antarctic minimum.

## 5.7 Tropospheric aerosol

Tropospheric aerosol is recognized as a major climate agent and is therefore commonly included in climate models, but there are important differences in the way it is implemented in different models. Table 32 of the supplement summarizes the way aerosol is represented in the models. Most models use a bulk aerosol formulation, i.e. aerosols are only externally mixed and are represented in size distributions with prescribed mean particle sizes. A few models do not include interactive aerosol at all but use some prescribed physical properties of aerosols, such as aerosol optical depth, surface area density, or mixing ratio (CNRM-CM5-3, EMAC, SOCOL). Others use more sophisticated methods of treating aerosol: A sectional approach does not assume a fixed size distribution and instead represents aerosol in size classes, with aerosol physics governing transitions between size classes. ULAQ uses this approach for almost all types of aerosol except nitrates (see below); MRI-ESM1 r1 treats sea salt in this way. A modal approach represents aerosols as log-normal distributions with both amplitude and mean radius becoming predicted quantities. This is considered intermediate in complexity between an expensive, physically consistent sectional and a computationally efficient, simple bulk approach. ULAQ is the only model to use this modal approach for nitrate aerosols.

CCSRNIES, CMAM, MOCAGE, TOMCAT, and UMSLIMCAT do not account for tropospheric aerosols at all.

## 5.8 Solar forcing

The recommended solar forcing data set contains daily solar irradiance, ionization rates by solar protons, and the geomagnetic activity index $A_p$ (http://solarisheppa.geomar.de/ccmi). Spectrally resolved solar irradiances for 1960-2010 were calculated



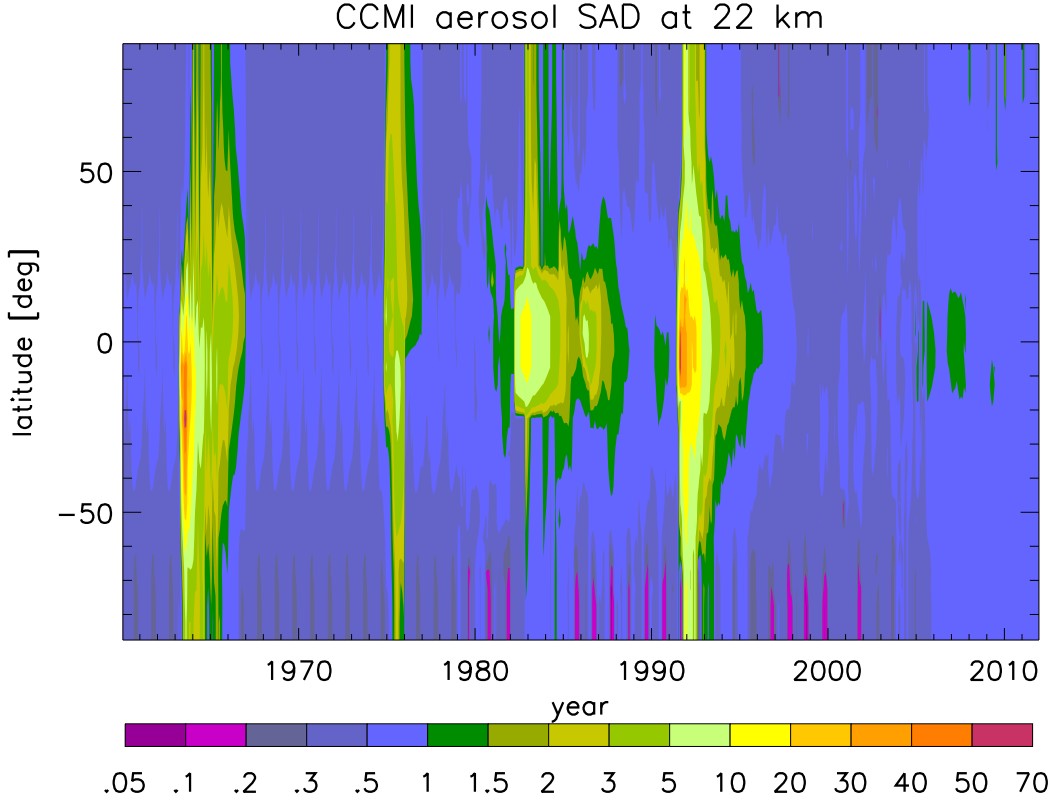

**Figure 4.** Zonal-mean aerosol surface area density ($\mu$m$^2$/cm$^3$) at 22 km. The discrete events are due to volcanic eruptions, superimposed on a much smaller non-volcanic background.

with the empirical NRLSSI model (Lean et al., 2005). The spectral grid width (1 nm bins from 0 to 750 nm, 5 nm bins from 750 nm to 5 $\mu$m, 10 nm bins from 5 to 10 $\mu$m, 50 nm bins from 10 to 100 $\mu$m) allows easy calculations of the spectral solar irradiance (SSI) for any specific model spectral grids, which should be applied to calculate the shortwave heating rates in the radiation module and the photolysis in the chemistry scheme. The ionization rates caused by solar protons for the same time

5   period are calculated using the Jackman et al. (2009) approach based on the proton flux measurements by several instruments onboard the GOES satellites. The recommended coefficients for the conversion of the ionization rates to in-situ HO$_x$ and NO$_x$ production intensity are also given by Jackman et al. (2009). For the models extending only to the mesopause, a time-varying geomagnetic activity index $A_p$ is provided as a proxy for the thermospheric NO$_x$ influx, which can be used to include indirect energetic particle effects using an approach similar to that defined by Baumgaertner et al. (2009). These data sets should be

10  applied in REF-C1 simulations covering 1960-2010. For SEN-C1-SSI simulations, the SATIRE SSI data set (Krivova et al., 2006) should be used instead of the NRLSSI data described above. This SSI data set exhibits larger UV variability which can have consequences not only for atmospheric heating but also for ozone chemistry (Ermolli et al., 2013). For REF-C2 simulations covering 2010-2100, it is recommended to repeat the SSI, SPE, and $A_p$ sequences of the last four solar activity



cycles (i.e., cycles 20-23). For the sensitivity experiment SEN-C2_SolarTrend covering 1960-2100 it is advised to introduce a declining trend in the solar activity, reflecting a widely discussed possible decline of solar activity in the future. The proposed trend is based on past solar activity cycles repeated in reverse order. Starting from 2011 it is recommended to apply daily SSI and particle output for the cycles 20, 19, 18, 17, 16, 15, 14, 13 and 12. The program to build daily future solar forcing

5  for different experiments is available from http://solarisheppa.geomar.de/ccmi. Figure 5 illustrates the $A_p$ index evolution for standard and sensitivity scenarios, showing a decline of the geomagnetic activity in the future, and the recommended solar irradiance for the 175-250 nm spectral band from its minimum value.

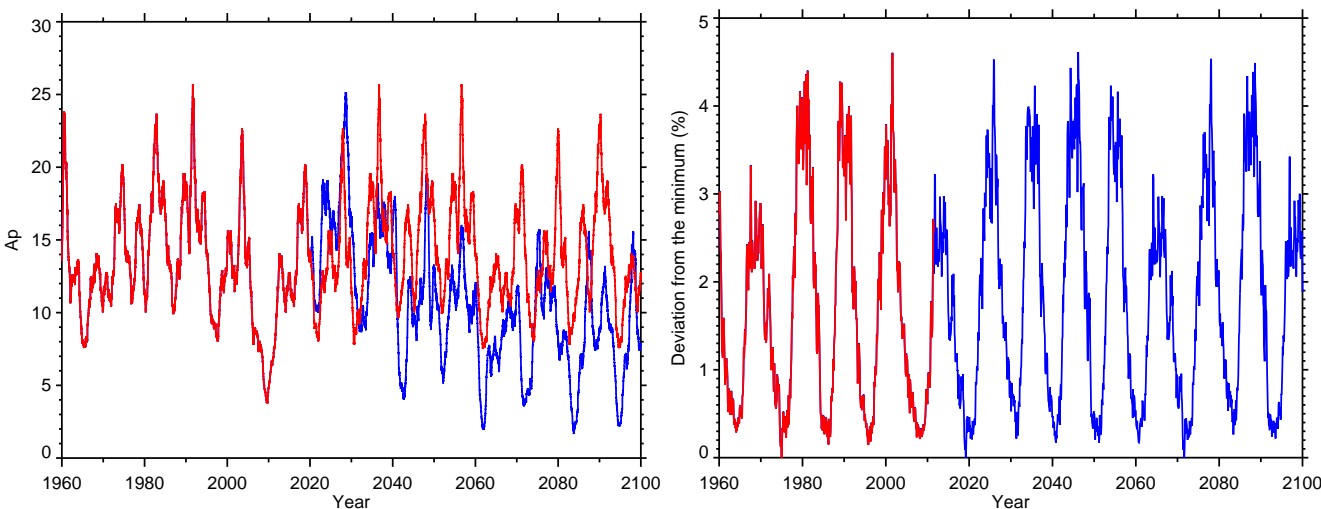

**Figure 5.** (left) Recommended $A_p$ index time series for SEN-C2_SolarTrend (blue) and the other reference and sensitivity simulations (red). The data are smoothed with 360-day wide window. (right) Deviation (%) of the recommended monthly mean solar irradiance for the 175-250 nm spectral band from its minimum value, to be used in all simulations. Red: Prior to and including 2010 the data are based on observations. Blue: Projected solar irradiance post 2010.

## 6  Availability of simulations

Tables 4-6 summarize the available simulations at the time of writing. As recommended, a large majority of models has

10  performed the reference simulations. A subset has produced REF-C1SD, reflecting that not all models have the capability to be nudged to meteorological fields. The sensitivity simulations, both SEN-C1 and SEN-C2, are less consistently covered, ranging from 1 to 12 simulations.

Most of the model output can be accessed via the British Atmospheric Data Center (BADC), which hosts the CCMI data archive. Some model institutions provide their data however directly via their local Earth System Federation Grid (ESFG) nodes

15  (see the list provided on http://www.met.reading.ac.uk/ccmi/?page_id=251). In some cases, the simulations are complete, but have not been fully uploaded for public access yet. In these cases, readers are advised to contact the corresponding model PIs.



## 7 Conclusions

The purpose of this paper has been to provide some overview information on the internal make-up of CCMI models, broadly characterize the forcings, and give an overview of available simulations under CCMI, mainly to inform other papers focussing on scientific results of CCMI. We have not assessed model performance, but it is clear from this paper that in the years since CCMVal-2 and ACCMIP, considerable progress has been made to improve the models' internal consistency, make them more physically based, and more comprehensive, on top of improving their resolutions. While these developments have to be welcomed, experience shows that simulations with a more physically consistent and comprehensive model, which is less constrained by external forcings, may not compare more favourably against observations than those produced by a more constrained model. This is particularly the case as Earth System Models increasingly cover aspects of the climate system that are challenging to capture numerically, such as atmospheric chemistry or biogeochemistry of the ocean. This complicates measuring progress in climate modelling and contributes to the perceived "failure" of the climate modelling community to narrow the range of climate futures produced in multi-model intercomparisons such as the 5th Coupled Model Intercomparison Project (CMIP5) or CCMI. Understanding how this diversity is linked to differences in model formulation can help explain such findings. The purpose of this paper, and a major motivation for CCMI, is to drive progress in this regard.

## 8 Code availability

Readers are advised to contact the model PIs to enquire about conditions of code availability for the 20 models documented in this paper.

## 9 Data availability

No model output was used in this paper. CCMI model data can be obtained from the CCMI data archive http://blogs.reading. ac.uk/ccmi/badc-data-access. Forcing data depicted in this paper are described and can be downloaded at http://blogs.reading. ac.uk/ccmi/reference-simulations-and-forcings.

## Appendix A: Individual model descriptions

### A1 ACCESS-CCM and NIWA-UKCA

NIWA-UKCA is a coupled atmosphere-ocean chemistry-climate model, based on the HadGEM3-AO model (revision 2) coupled to the NIWA-UKCA gas-phase chemistry scheme. It is identical to ACCESS-CCM, except that ACCESS-CCM uses prescribed sea-surface conditions in all simulations. Relative to the UMUKCA models used for CCMVal-2, both models now feature a medium-complexity tropospheric hydrocarbon oxidation scheme, including the Mainz Isoprene Mechanism (Pöschl et al., 2000) and the FAST-JX online photolysis scheme (Telford et al., 2013). NIWA-UKCA uses an interactive ocean and sea



ice module (Hewitt et al., 2011). In transitioning to HadGEM3, atmospheric physics was updated; in particular, the models now use the PC2 cloud scheme (Wilson et al., 2008). The models are run at a resolution of N48L60 ($3.75° \times 2.5°$) in the atmosphere and (for NIWA-UKCA) $\sim 2°$ and 31 levels in the ocean.

## A2  CCSRNIES-MIROC3.2

CCSRNIES-MIROC3.2 CCM was constructed on the basis of the MIROC3.2 general circulation model (GCM), which was used for future climate projection in the fourth and fifth assessment reports of the Intergovernmental Panel on Climate Change (IPCC, 2007, 2013). The updated CCM introduces the stratospheric chemistry module of CCSRNIES CCM that was used for CCMVal and CCMVal-2. CCSRNIES-MIROC3.2 CCM has a new higher resolution radiation scheme for the spectral bins (32 bins) than that of CCSRNIES CCM (18 bins). The new CCM uses a semi-Lagrangian scheme for tracer transport,

whilst CCSRNIES CCM used a spectral transport scheme. The new CCM is not coupled with the ocean; SST and sea ice are prescribed in the simulations.

## A3  CESM1 CAM4-chem and CESM1 WACCM

The Community Earth System Model, version 1 (CESM1) is a coupled climate model for simulating the Earth' climate system. The atmospheric component is the Community Atmosphere Model, version 4 (CAM4) (Neale et al., 2013), which uses

a finite volume dynamical core (Lin et al., 2004) for the tracer advection. Two versions of CAM4 participated in CCMI: 1) A lower lid model reaching up to about 40km altitude (CESM1 CAM4-chem); 2) and a high top model that extends to approximately 140km altitude (Whole Atmosphere Community Climate Model Version 4, CESM1 WACCM4). The horizontal resolution used for all CCMI simulations is $1.9° \times 2.5°$ (latitude $\times$ longitude). Both model versions include detailed and identical representation of tropospheric and stratospheric (TS) chemistry and interactive tropospheric aerosols (Tilmes et al.,

2016). The polar heterogeneous chemistry was recently updated (Wegner et al., 2013) and further evaluated by Solomon et al. (2015). CESM1 WACCM also includes a representation of physics and chemistry of the mesosphere lower thermosphere (MLT) region (Marsh et al., 2013). The TS (CESM1 CAM4-chem) and TSMLT (CESM1 WACCM4) chemical mechanisms include 171 and 183 species respectively contained within the $O_x$, $NO_x$, $HO_x$, $ClO_x$, $BrO_x$, and $FO_x$ chemical families, along with $CH_4$ and its degradation products. In addition, 17 primary non–methane hydrocarbons and related oxygenated organic

compounds are included. All CCMI scenarios use the same TS and TSMLT chemical mechanism. The previous CCMVal2 version of CESM1-WACCM simulated Southern Hemisphere winter and spring temperatures that were too cold compared with observations. Among other consequences, with the recent updates to the heterogeneous chemistry module, this "cold pole bias" leads to unrealistically low ozone column amounts in Antarctic spring. In all CCMI simulations, the cold pole problem is addressed by introducing additional mechanical forcing of the circulation via parameterized gravity waves (Garcia et al.,

30   2016).



## A4   CHASER (MIROC-ESM)

The CHASER model (Sudo et al., 2002; Sudo and Akimoto, 2007), developed mainly at Nagoya University and the Japan Agency for Marine-Earth Science and Technology (JAMSTEC), is a coupled chemistry-climate model, simulating atmospheric chemistry and aerosols. Aerosols are handled by the SPRINTARS module (Takemura et al., 2005). It has been developed also

in the framework of the MIROC Earth System Model, MIROC-ESM-CHEM (Watanabe et al., 2010). CHASER simulates detailed chemistry in the troposphere and stratosphere with an on-line aerosol simulation including production of particulate nitrate and secondary organic aerosols. For this study, the model's horizontal resolution is selected to be T42 ($2.8° \times 2.8°$) with 57 layers in the vertical extending from the surface up to about 55 km altitude. As for the overall model structure, CHASER is fully coupled with the climate model core MIROC, permitting atmospheric constituents (both gases and aerosols) to interact

radiatively and hydrologically with meteorological fields in the model. The chemistry component of CHASER considers the $O_x$-$NO_x$-$HO_x$-$CH_4$-CO chemical system with oxidation of NMVOCs, halogen chemistry, and the $NH_x$-$SO_x$-$NO_3$ system. In total 96 chemical species and 287 chemical reactions are considered. In the model, primary NMVOCs include $C_2H_6$, $C_2H_4$, $C_3H_8$, $C_3H_6$, $C_4H_{10}$, acetone, methanol, and biogenic VOCs (isoprene, terpenes).

## A5   CMAM

Compared with the model version used for CCMVal-2, the CMAM used for the CCMI simulations calculates chemistry throughout the troposphere, though the only hydrocarbon considered is methane. While CMAM was interactively coupled to an ocean model for CCMVal-2, specified SSTs and sea-ice fields were used for all CCMI simulations. The horizontal resolution has increased from T31 to T47 and while spectral advection is still used for all chemical tracers, for $HNO_3$ and $NO_x$ a logarithmic transformation of the mixing ratio (Scinocca et al., 2008) is advected to better preserve the strong horizontal

gradients in the troposphere. For CCMVal-2 a constant dry deposition velocity was used to provide a tropospheric sink for selected species; here wet deposition is calculated interactively with the stratiform / deep convection parameterizations and dry deposition uses a 'big-leaf' approach that is tied to the model land surface scheme. The look-up table for photolysis rates has been expanded to take account of surface albedo and a correction to the clear-sky rates is made for clouds following the approach of Chang et al. (1987). Hydrolysis of $N_2O_5$ in the troposphere has been included, using a monthly-varying clima-

tology of sulfate aerosols from a more recent version of the Canadian climate model (von Salzen et al., 2013) and reaction probabilities of Davis et al. (2008) assuming ammonium sulfate.

## A6   CNRM-CM5-3

The CNRM-CM5-3 chemistry-climate model, is based on the CNRM-CM5-3 AOGCM of CNRM/CERFACS whose version 5.1 has been used in CMIP5 simulations and is described by Voldoire et al. (2012). The CCM includes some fundamental

changes from the previous version (CNRM-ACM) which was extensively evaluated in the context of the CCMVal-2 validation activity. The most notable changes concern the radiative code of the GCM (Morcrette, 1990, 1991; Morcrette et al., 2001), the parametrization of non-orographic gravity waves, stochastic parametrization triggered by convection as described by (Lott



and Guez, 2013), and the inclusion of the detailed stratospheric chemistry on-line within the GCM (Michou et al., 2011). To clarify, CCMI simulations have been performed in an AMIP type mode, the atmospheric GGM (v6.03) being forced by SSTs and sea ice, without the use of the SURFEX external surface scheme.

## A7 EMAC

The Modular Earth Submodel System (Jöckel et al., 2005, 2006, 2010) is a software package providing a framework for a standardised, bottom-up implementation of Earth System Models with flexible complexity. "Bottom-up" means, the MESSy software provides an infrastructure with generalised interfaces for the standardised control and interconnection (coupling) of low-level ESM components (i.e., dynamic cores, physical parameterisations, chemistry packages, diagnostics, etc.), which are called submodels. MESSy comprises currently about 60 submodels (i.e., coded according to the MESSy standards) in different categories: infrastructure (i.e., framework) submodels, atmospheric chemistry related submodels, physics related submodels, and diagnostic submodels. The ECHAM/MESSy Atmospheric Chemistry (EMAC) model uses the Modular Earth Submodel System to link multi-institutional computer codes to the core atmospheric model, i.e., the 5th generation European Centre Hamburg general circulation model (Roeckner et al., 2003, 2006). Updates used for CCMI (EMAC version 2.51) are documented in detail by Jöckel et al. (2016).

## A8 GEOSCCM

The Goddard Earth Observing System Chemistry-Climate Model (GEOSCCM) is based on GEOS-5 GCM (Molod et al., 2012, 2015) coupled to the stratospheric and tropospheric (StratTrop) Global Modeling Initiative (GMI) chemical mechanism (Strahan et al., 2007; Duncan et al., 2007). This version uses a C48 cubed sphere grid, which has been regridded to $2.5°$ longitude $\times 2°$ latitude horizontal resolution and 72 vertical layers up to 80 km. The response of tropospheric ozone to variations in the El Nino Southern Oscillation (ENSO) compared to observations were described by Oman et al. (2011, 2013). An earlier version of the model contributed to the ACCMIP activity (Lamarque et al., 2013).

## A9 GFDL-AM3 AND GFDL-CM3

AM3 is the atmospheric component of the Geophysical Fluid Dynamic Laboratory (GFDL) global coupled atmosphere-ocean-land-sea ice model (CM3), which includes interactive stratosphere-troposphere chemistry and aerosols at C48 cubed-sphere horizontal resolution (approximately $2° \times 2°$) (Donner et al., 2011; Austin et al., 2013; Naik et al., 2013). In support of CCMI, we conduct a suite of multi-decadal hindcast simulations (1979-2014) designed to isolate the response of atmospheric constituents to historical changes in human-induced emissions, methane, wildfires, and meteorology. Details of these simulations are described by Lin et al. (2014, 2015a, b). We implement a height-dependent nudging technique, relaxing the model to NCEP $u$ and $v$ with a time scale of 6 h in the surface level, but weakening the nudging strength linearly with decreasing pressure (e.g., relaxing with a time scale of 60 h by 100 hPa and 600 h by 10 hPa) (Lin et al., 2012a). To quantify stratospheric influence on tropospheric ozone, we define a stratospheric ozone tracer relative to a dynamically varying tropopause and subjecting it to



chemical and depositional loss in the same manner as odd oxygen in the troposphere (Lin et al., 2012b, 2015a). These AM3 simulations have been evaluated against a broad suite of observations. Analysis of satellite measurements, daily ozonesondes, and multi-decadal in-situ observation records indicates that the nudged GFDL-AM3 model captures many salient features of observed ozone over the North Pacific and North America, including the influences from Asian pollution events (Lin et al.,

2012a), deep tropopause folds (Lin et al., 2012b), as well as their variability on interannual to decadal time scales (Lin et al., 2014, 2015a) and long-term trends (Lin et al., 2015b). The model also captures interannual variability of ozone in the lower stratosphere and its response to ENSO events and to volcanic aerosols as measured by ozonesondes (Lin et al., 2015a).

## A10 HadGEM3-ES

The Met Office model (HadGEM3-ES, formerly UMUKCA-METO) has changed significantly since CCMVal-2. The under-

lying atmosphere model is now HadGEM3 (Walters et al., 2014), with horizontal resolution increased from $3.75°$ longitude $\times 2.5°$ latitude to $1.875°$ longitude $\times 1.25°$ latitude, and the number of vertical levels spanning the model domain 0-85 km increased from 60 to 85. The move to the HadGEM3 model has significantly reduced two critical biases seen in UMUKCA-METO simulations in which stratospheric air was too old and the tropical tropopause too warm (Morgenstern et al., 2009). As a consequence of the improvements in tropical tropopause temperatures, water vapour concentrations entering the strato-

sphere are no longer prescribed and are now interactively determined by the model. For the scenario simulations coupled ocean (NEMO vn3.4; Madec, 2008) and sea ice (CICE vn4.1; Hunke and Lipscombe, 2008) models are now included. Significant developments to the UKCA chemistry component (Morgenstern et al., 2009; O'Connor et al., 2014) include the replacement of the stratosphere-only scheme used in UMUKCA-METO with a combined stratosphere-troposphere chemistry scheme, with increased numbers of tracers, chemical species and reactions, interactive lightning emissions (O'Connor et al., 2014), interac-

tive photolysis rates (Fast-JX; Telford et al., 2013), the CLASSIC aerosol scheme (Bellouin et al., 2011), and a resistance-type approach to dry deposition (O'Connor et al., 2014).

Changes since involvement in the ACCMIP project include: Replacement of the atmosphere model from HadGEM2 model, with 38 vertical levels, to HadGEM3, with 85 vertical levels. The troposphere-only chemistry scheme was replaced by a combined stratosphere-troposphere chemistry scheme. The Mainz Isoprene Mechanism (MIM; Pöschl et al. (2000)) is included

and the offline photolysis scheme has been replaced with Fast-JX (Telford et al., 2013).

## A11 LMDz-REPROBUS

LMDz-REPROBUS is a coupled chemistry-climate model, formed by the coupling of the LMD GCM and the REPROBUS atmospheric chemistry module. When linked to the NEMO ocean model, the configuration is identical to the IPSL atmosphere-ocean climate model but with atmospheric chemistry. The version LMDZ-REPROBUS model used initially for CCMVal-2 had

50 levels and a resolution of $2.5°$ latitude $\times 3.75°$ longitude with a top at about 65 km. The CCMI simulations already completed have been performed with the CMIP5 version (LMDz-REPROBUS-CM5) that have 39 levels and a resolution of $2.5°$ latitude $\times 3.75°$ longitude with a top at about 70 km (Dufresne et al., 2013). We plan to rerun the same CCMI simulations with the CM6 version that has 79 levels and a resolution of $1.25°$ latitude $\times 2.5°$ longitude with a top at about 80 km.



## A12   MOCAGE

MOCAGE (Modèle de Chimie Atmosphérique de Grande Echelle) is Météo-France's Chemical Transport Model. MOCAGE combines the RACM (Stockwell et al., 1997) tropospheric and the REPROBUS (Lefèvre et al., 1994) stratospheric chemistry schemes, consistently applied from the surface to the model top. It simulates 109 gaseous species (there are no aerosols in these CCMI runs), that are grouped in families, with 91 being transported. In the stratosphere, 9 heterogeneous reactions are described, using the parameterization of Carslaw et al. (1995a). Moreover, 52 photolysis and 312 thermal reactions are described. The photolysis rates follow look-up tables and are modified to account for cloudiness, following Chang et al. (1987). The model includes a reaction pathway for $HO_2 + NO$ to yield $HNO_3$ (Butkovskaya et al., 2007).

The resolution of the model is $2° \times 2°$ on a latitude-longitude grid, with 47 levels to 5 hPa. For the REF-C1SD and SEN-C1SD-fEmis experiments, we use ERA-Interim forcings. For the REF-C1 and REF-C2 experiments, the meteorological forcing is taken from an update of the CNRM-CM model, which was used for CMIP5 simulations (Voldoire et al., 2012). However, convective transport of species is recomputed following the parameterization by Bechtold et al. (2001). Convective in-cloud scavenging is determined in the updraft (Mari et al., 2000), whereas wet deposition due to stratiform precipitations follow Giorgi and Chameides (1986). A one-year simulation has been performed to compute dry deposition velocities following Wesely's approach. Values have been averaged to get monthly diurnal profiles. The same values have been used for all simulations.

Except lightning $NO_x$, natural emissions are monthly mean distributions taken from GEIA inventories. Lightning $NO_x$ is parameterized in the convection scheme following Price et al. (1997) and is hence climate-sensitive. Methane concentrations were prescribed at the surface following a monthly zonal climatology taking the evolution of the global value as a function of RCPs into account.

## A13   MRI-ESM1r1

MRI-ESM1r1 is an updated version of the Earth System Model MRI-ESM1 which was used for future climate projection in the 5th Assessment Report of the Intergovernmental Panel on Climate Change (IPCC, 2013). The vertical resolution of MRI-ESM1r1 (L80) is improved compared to MRI-ESM1 (L48). The SCUP coupler (Yoshimura and Yukimoto, 2008) is used to couple the atmosphere, ocean, aerosol, and (gas-phase) chemistry modules which make up MRI-ESM1r1. The chemistry module is MRI-CCM2 (Deushi and Shibata, 2011), which is an updated version of MRI-CCM1 used for CCMVal-2. In MRI-CCM2, a tropospheric hydrocarbon oxidation scheme with medium-complexity is newly added.

## A14   SOCOLv3

Since CCMVal-2 the SOCOL model (Solar-Climate-Ozone Links, Stenke et al., 2013) has significantly changed. SOCOL v2, which participated in CCMVal-2, was a combination of the GCM MA-ECHAM4 (Manzini et al., 1997) and the CTM MEZON (Rozanov et al., 1999; Egorova et al., 2003), while the third and current version, SOCOLv3, is based on MA-ECHAM5 (Roeckner et al., 2006; Manzini et al., 2006). The advection of chemical trace species is now calculated by the flux-form semi-Lagrangian scheme of Lin and Rood (1996) instead of the previously applied hybrid advection scheme. This change made the





mass correction applied to certain tracers in SOCOLv2 obsolete. Furthermore, the unsatisfying separation of tropospheric and stratospheric water vapor fields in SOCOLv2 has also become obsolete. SOCOLv3 considers only one water vapor field, i.e., the ECHAM5 water vapor. Advection, convection and the tropospheric hydrological cycle are calculated by the GCM, while chemical water vapor production/destruction as well as PSC formation are calculated by the chemistry module.

For CCMI SOCOL was run with T42 horizontal resolution, which corresponds approximately to $2.8° \times 2.8°$, and with 39 vertical levels between the Earth' surface and 0.01 hPa ($\sim 80$ km). Further important modifications for the CCMI set up include an isoprene oxidation mechanism (Pöschl et al., 2000), the online calculation of lightning $NO_x$ emissions (Price and Rind, 1992), treatment of the effects produced by different energetic particles (Rozanov et al., 2012), updated reaction rates and absorption cross sections (Sander et al., 2011b), improved solar heating rates (Sukhodolov et al., 2014), as well

as a parameterisation of cloud effects on photolysis rates (Chang et al., 1987). Furthermore, the ODS species are no longer transported as families, but as separate tracers.

## A15   TOMCAT

TOMCAT is a global 3-D off-line chemical transport model (Chipperfield, 2006). The model is usually forced by ECMWF meteorological (re)analyses, although GCM output can also be used. When using ECMWF fields, as in the CCMI experiments,

the model reads in the 6-hourly fields of temperature, humidity, vorticity, divergence and surface pressure. The resolved vertical motion is calculated online from the vorticity. The model has parameterisations for sub-gridscale tracer transport by convection (Stockwell and Chipperfield, 1999; Feng et al., 2011) and boundary layer mixing (Holtslag and Boville, 1993). Tracer advection is performed using the conservation of second order moments scheme by Prather (1986). The CTM be used with a variety of chemistry and aerosol schemes including stratospheric chemistry (Chipperfield et al., 2015), tropospheric chemistry (e.g.,

Monks et al., 2012) and idealised tracers. For the CCMI experiments the model was run at horizontal resolution of $2.8° \times 2.8°$ with 60 levels from the surface to $\sim 60$ km. Experiments with stratospheric chemistry and idealised tracers were performed.

## A16   ULAQ-CCM

The ULAQ-CCM is a climate-chemistry coupled model with an interactive aerosol module (a compact description was given by Morgenstern et al., 2010, for SPARC-CCMVal-2). Since then, some updates have been made to the model (Pitari et al., 2014):

(a) increase in horizontal and vertical resolution; (b) inclusion of a numerical code for the formation of upper tropospheric cirrus cloud ice particles (Kärcher and Lohmann, 2002; Pitari et al., 2015a); (c) upgrade of the radiative transfer code for calculations of photolysis, heating rates and radiative forcing. This is a two-stream $\delta$-Eddington approximation model operating on-line in the ULAQ-CCM and used for chemical species photolysis rate calculation at UV-visible wavelengths and for solar heating rates and radiative forcing at UV-VIS-NIR bands (Randles et al., 2013; Pitari et al., 2015b). In addition, a companion

broadband, $k$-distribution longwave radiative module is used to compute radiative transfer and heating rates in the planetary infrared spectrum (Chou et al., 2001; Pitari et al., 2015c). Calculations of photolysis rates and radiative fluxes have been evaluated in the framework of SPARC CCMVal (SPARC, 2013) and AeroCom inter-comparison campaigns (Randles et al., 2013). The chemistry-aerosol module is organized with all medium and short-lived species grouped in families. It includes



the major components of tropospheric aerosols (sulfate, carbonaceous, soil dust, sea salt), with calculation at each size bin of surface fluxes, removal and transport terms, in external mixing conditions. A modal approximation is used for nitrate aerosols. Wet and dry depositions are treated following Müller and Brasseur (1995), using a climatological cloud distribution. Lower stratospheric denitrification and dehydration are calculated using the predicted size distribution of PSC particles.

## A17   UMSLIMCAT

The UMSLIMCAT has only undergone minor changes since CCMVal-2. The model is based on a old version of the Met Office Unified Model and, although it performs well in terms of stratospheric chemistry and dynamics, the model is not actively developed. Core UMSLIMCAT simulations are performed in order to increase the range of simulations available and to provide some continuity with previous CCM studies. Compared to CCMVal-2 the minor model updates are: (i) Updated photolysis scheme with an improved treatment of ozone profiles in the on-line look-up table, (ii) the use of the CCMI aerosol surface area density (SAD) and (iii) an updated solar flux representation. Dhomse et al. (2011, 2013, 2015) describe the implementation of this representation and present an analysis of solar flux variability and volcanic aerosol in the model.

## A18   UMUKCA-UCAM

UMUKCA-UCAM is an atmosphere only chemistry-climate model, based on the HadGEM3 model (revision 2). The chemistry in UMUKCA-UCAM is based on a similar scheme as was used in the UMUKCA models in CCMVal2 (focusing on the chemistry of stratosphere, Bednarz et al., 2016), but with an explicit treatment of halogen source gases, i.e. no lumping. Since CCMVal2, significant improvements to the model physics have been made and except that the model resolution is degraded to run at N48L60 ($3.75° \times 2.5°$) in the atmosphere, the model physics is identical to HadGEM3. Relative to the UMUKCA models used for CCMVal-2, the FAST-JX online photolysis scheme (Telford et al., 2013) is now included, as is interactive lightning emissions, the CLASSIC aerosol scheme (Bellouin et al., 2011), and a resistance-type approach to dry deposition.

### Appendix B:   Deviations from CCMI recommendations

We list here the ways in which simulations and model set-ups deviate from Eyring et al. (2013). Also, simulations submitted to the archive that are additional to those solicited by Eyring et al. (2013) and Hegglin et al. (2016) are described here.

### B1   ACCESS CCM and NIWA-UKCA

For some simulations, anthropogenic NMVOC emissions were held at their 1960 levels for 1960-2000 in about half of the NIWA-UKCA simulations. This error was picked up and corrected for the later simulations but remains in earlier simulations. Simulations affected by this problem include: REF-C1 (r2, r3), REF-C2 (r1,r2,r3,r4), SEN-C2-fODS (r1), and SEN-C2-fGHG (r1). Not affected are REF-C1 (r1), SEN-C1-fEMIS (r1), REF-C2 (r5), SEN-C2-fODS (r2), SEN-C2-fGHG (r2, r3), **SEN-C2-fCH4 (r1), and SEN-C2-fN2O (r1)**.



As noted before, ACCESS CCM and NIWA-UKCA do not consider the radiative impacts of stratospheric aerosol. Also there is no variance correction applied to sea surface temperatures in the simulations without interactive ocean.

## B2   CCSRNIES-MIROC3.2

HadISST1 data were used for REF-C1 and REF-C1SD simulations. Chemical reactions important in the troposphere are not
included, but the stratospheric chemistry scheme is just used in the troposphere. Solar radiation at wavelengths shorter than 177.5 nm is not considered except for Lyman-$\alpha$. Atmospheric ionization by solar protons is not included.

## B3   CMAM

The ACCMIP historical database of emissions (Lamarque et al., 2010) was used for the REF-C1 and REF-C1SD simulations up to the year 2000, with the RCP8.5 emissions used for the following years. It was also used up to 2000 for the REF-C2
and associated scenario simulations. Emissions at intermediate years were linearly interpolated from the years given in the database. An additional emission of CO of 250 Tg(CO) per year was included to account for CO from isoprene oxidation, with the emissions distributed following the monthly emissions of isoprene from Guenther et al. (1995). No variance correction was applied to the specified SSTs.

## B4   EMAC

Due to a unit conversion error at data import, the extinction of stratospheric aerosols was too low, by a factor of approximately 500. The effect of stratospheric background aerosol on radiative heating rates has been tested by sensitivity simulations and estimated to be smaller than the interannual standard deviation. However, the dynamical effects of large volcanic eruptions (e.g. Mt. Pinatubo in 1991, El Chichón in 1982, etc.) are essentially not represented in the simulations, except for the contribution to the tropospheric temperature signal induced by the prescribed SSTs. The chemical effects (through heterogeneous chemistry),
however, are included, since the prescribed aerosol surface areas were treated correctly.

Next, due to an error in the model setup, the timing of the road traffic emissions was unfortunately wrong: instead of updating the monthly input fields every month, they have been updated only every year, thus in 1950 emissions of January 1950 have been used, in 1951 the emissions of February 1950, etc.

And last, but not least, some of the diagnostic tracers have been treated differently.

More details on the deviations from the CCMI recommendations are documented by Jöckel et al. (2016, see their section 3.12 and Table A1).

## B5   GFDL-AM3

The AM3_BASE simulation (i.e., REF-C1SD) applies interannually-varying emissions of aerosol and ozone precursors from human activity, based on Lamarque et al. (2010) for 1980-2000 and RCP 8.5 projections (Riahi et al., 2011) beyond 2005,
linearly interpolated for intermediate years. The AM3_FIXEMIS simulation (i.e., SEN-C1SD-fEmis) with anthropogenic and





biomass burning emissions set to the 1970-2010 climatology and methane held constant at 2000 levels, is designed to isolate the role of meteorology. The IAVFIRE simulation (i.e., SEN-C1SD-Emis) applies interannual-varying monthly mean emissions from biomass burning based on Schultz et al. (2008) for 1970 to 1996 and GFEDv3 (Van der Werf et al., 2010) for 1997-2010. Otherwise, all forcings are the same as in FIXEMIS. The BASE, FIXEMIS, and IAVFIRE simulations with modified emissions

are nudged to NCEP reanalysis winds over 1980 to 2010. We also conduct four ensemble simulations without nudging, driven by prescribed sea surface temperatures (SSTs) and atmospheric radiative forcing agents over 1960 to 2010 (SEN-C1-Emis; Table 7). In SEN-C1SD-fEmis, emissions of ozone and aerosol precursors are fixed to the 1970-2010 climatology, instead of the 1980 levels recommended by CCMI. Due to an error in data processing, anthropogenic emissions of aerosol precursors ($SO_2$, BC, and OC) after 1996 in the REF-C1SD simulation do not follow the CCMI recommendation. Otherwise denoted in

Table 7, all forcings follow the CCMI recommendations.

## B6 HadGEM3-ES

The specified dynamics simulation (REF-C1SD) uses an anomaly correction to the ERA-I forcing data, as outlined in Mclandress et al. (2014). Two REF-C2 simulations only start in 2000. The three SEN-C2-fGHG simulations are forced with fixed year-2000 GHG mixing ratios not 1960 ones, and also only start in 2000.

## B7 MRI-ESM1r1

The molecular weight of sulfate aerosols ($SO_4$) due to volcanic eruptions was inappropriately set to that of sulfur atom (S) in our REF-C1 and REF-C1SD simulations. In result, the amount of volcanic aerosol in these simulations was one-third of its correct amount. Molecular weights of other aerosols (i.e. anthropogenic, biogenic, dust, etc.) are appropriately treated.

## B8 SOCOLv3

Sea surface temperatures for REF-C2 and all sensitivity simulations based on REF-C2 were taken from the CESM1-CAM5 model.

## B9 ULAQ CCM

All CCMI experiments have been conducted following the CCMI recommendations. For the sensitivity cases SEN-C2-fGHG, SEN-C2-fODS, and SEN-C2-fODS2000, the following procedure has been used for $CH_4$, $N_2O$ and CFCs, which are both

GHG and ODS. These species were fixed in the radiation-dynamics-climate modules in the fGHG experiment, leaving them to evolve in time for chemistry. The opposite choice was made for the two fODS experiments (1960, 2000), i.e., fixing these species in chemistry and letting them evolve in the radiation-dynamics-climate modules.





## B10   UMUKCA-UCAM

The stratospheric aerosol climatology used is SPARC (2006), and is included in the chemistry, photolysis, and radiation schemes. Surface emissions (of $NO_x$, CO, and HCHO) and the $NO_x$ aircraft emissions are the same as used in the CCM-Val2 REF-B2 simulation.

*Author contributions.*  OM and MIH have devised the concept and written most of the paper. The other authors have contributed information pertaining to their individual models and have revised and helped formulate the paper.

*Acknowledgements.*  We thank the Centre for Environmental Data Analysis (CEDA) for hosting the CCMI data archive. This work has been supported by NIWA as part of its Government-funded, core research. OM acknowledges support from the Royal Society Marsden Fund, grant 12-NIW-006. The authors wish to acknowledge the contribution of NeSI high-performance computing facilities to the results of this research.

New Zealand's national facilities are provided by the New Zealand eScience Infrastructure (NeSI) and funded jointly by NeSI's collaborator institutions and through the Ministry of Business, Innovation & Employment's Research Infrastructure programme (https://www.nesi.org.nz). The SOCOL team acknowledges support from the Swiss National Science Foundation under grant agreement CRSII2_147659 (FUPSOL II). CCSRNIES' research was supported by the Environment Research and Technology Development Fund (2-1303) of the Ministry of the Environment, Japan, and computations were performed on NEC-SX9/A(ECO) computers at the CGER, NIES. Wuhu Feng (NCAS)

provided support for the TOMCAT simulations. NB, SCH, and FMO'C and the development of HadGEM3-ES were supported by the Joint UK DECC/Defra Met Office Hadley Centre Climate Programme (GA01101). NB and SCH also acknowledge additional support from the European Project 603557-STRATOCLIM under the FP7-ENV.2013.6.1-2 programme. FMO'C acknowledges additional support from the Horizon 2020 European Union's Framework Programme for Research and Innovation CRESCENDO project under Grant Agreement No. 641816. SB acknowledges support from the European Project 603557-STRATOCLIM under the FP7-ENV.2013.6.1-2 programme and from

the Centre National d'Etudes Spatiales (CNES, France) within the SOLSPEC project. KS and RS acknowledge funding from the Australian Government's Australian Antarctic science grant program (FoRCES 4012), the Australian Research Council's Centre of Excellence for Climate System Science (CE110001028), the Commonwealth Department of the Environment (grant 2011/16853) and computational support from National computational infrastructure INCMAS project q90. The CNRM-CM chemistry-climate people acknowledge the support from Météo-France, CNRS, and CERFACS, and in particular the work of the entire team in charge of the CNRM/CERFACS climate model.



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





**Table 1.** Participating models and contact information. For abbreviations of institution names, see the authors' affiliations.

| Model names | Institutions | Investigators | Email addresses |
|---|---|---|---|
| ACCESS CCM | U. Melbourne, AAD, NIWA | K. Stone, R. Schofield, A. Klekociuk, D. Karoly, O. Morgenstern | stonek@mit.edu<br>robyn.schofield@unimelb.edu.au<br>andrew.klekociuk@aad.edu.au |
| CCSRNIES MIROC3.2 | NIES | H. Akiyoshi, Y. Yamashita | hakiyosi@nies.go.jp<br>yyousuke@jamstec.go.jp |
| CESM1 CAM4-chem | NCAR | S. Tilmes, J.-F. Lamarque | tilmes@ucar.edu<br>lamar@ucar.edu |
| CESM1 WACCM | NCAR | D. Kinnison, R. R. Garcia, A. K. Smith, A. Gettelman, D. Marsh, C. Bardeen, M. Mills | dkin@ucar.edu, rgarcia@ucar.edu<br>aksmith@ucar.edu, andrew@ucar.edu<br>marsh@ucar.edu, bardeenc@ucar.edu<br>mmills@ucar.edu |
| CHASER (MIROC-ESM) | U. Nagoya, JAMSTEC, NIES | K. Sudo, T. Nagashima | kengo@nagoya-u.jp<br>nagashima.tatsuya@nies.go.jp |
| CMAM | CCCma, Environment and Climate Change Canada | D. Plummer, J. Scinocca | david.plummer@canada.ca<br>john.scinocca@canada.ca |
| CNRM-CM5-3 | CNRM Météo-France CNRS - CERFACS | M. Michou, D. Saint-Martin | martine.michou@meteo.fr<br>david.saint-martin@meteo.fr |
| EMAC | DLR-IPA, KIT-IMK-ASF, KIT-SCC-SLC, FZJ-IEK-7, FUB, UMZ-IPA, MPIC, CYI | P. Jöckel, H. Tost, A. Pozzer, M. Kunze, O. Kirner, S. Brinkop, D. S. Cai, J. Eckstein, F. Frank, H. Garny, K.-D. Gottschaldt, P. Graf, V. Grewe, A. Kerkweg, B. Kern, S. Matthes, M. Mertens, S. Meul, M. Nützel, S. Oberländer-Hayn, R. Ruhnke, R. Sander | messy_admin@lists.mpic.de<br>patrick.joeckel@dlr.de |
| GEOSCCM | NASA/GSFC | L. D. Oman, S. E. Strahan | luke.d.oman@nasa.gov<br>susan.e.strahan@nasa.gov |
| GFDL-AM3/CM3 | NOAA GFDL | M. Y. Lin, L. W. Horowitz | meiyun.lin@noaa.gov<br>larry.horowitz@noaa.gov |
| HadGEM3-ES | MOHC | F. M. O'Connor, N. Butchart, S. C. Hardiman, S. T. Rumbold | fiona.oconnor@metoffice.gov.uk<br>neal.butchart@metoffice.gov.uk |
| LMDz-REPROBUS | IPSL | S. Bekki , M. Marchand, F. Lott, D. Cugnet, L. Guez, F. Lefevre, S. Szopa | slimane@latmos.ipsl.fr<br>marion.marchand@latmos.ipsl.fr |
| MOCAGE | Météo France CNRS | B. Josse, V. Marecal | beatrice.josse@meteo.fr<br>virginie.marecal@meteo.fr |
| MRI-ESM1r1 | MRI | M. Deushi, T. Y. Tanaka, K. Yoshida | mdeushi@mri-jma.go.jp<br>yatanaka@mri-jma.go.jp<br>kyoshida@mri-jma.go.jp |
| NIWA-UKCA | NIWA | O. Morgenstern, G. Zeng | olaf.morgenstern@niwa.co.nz<br>guang.zeng@niwa.co.nz |
| SOCOL | PMOD/WRC, IAC/ETHZ | E. Rozanov, A. Stenke, L. Revell | eugene.rozanov@pmodwrc.ch<br>andrea.stenke@env.ethz.ch<br>laura.revell@env.ethz.ch |
| TOMCAT | U. Leeds | S. Dhomse, M. P. Chipperfield | m.chipperfield@leeds.ac.uk<br>s.s.dhomse@leeds.ac.uk |
| ULAQ-CCM | U. L'Aquila | G. Pitari, E. Mancini, G. Di Genova | gianni.pitari@aquila.infn.it<br>eva.mancini@aquila.infn.it<br>glauco.digenova@aquila.infn.it |
| UMSLIMCAT | U. Leeds | S. Dhomse, M. P. Chipperfield | m.chipperfield@leeds.ac.uk<br>s.s.dhomse@leeds.ac.uk |
| UMUKCA-UCAM | U. Cambridge | N. L. Abraham, A. T. Archibald, R. Currie, J. A. Pyle | luke.abraham@atm.ch.cam.ac.uk<br>ata27@cam.ac.uk<br>rc454@cam.ac.uk<br>john.pyle@atm.ch.cam.ac.uk |





**Table 2.** Model versions and key references.

| Model | Revision/version | Reference(s) | CCMVal-2 precursor model |
|---|---|---|---|
| ACCESS CCM, NIWA-UKCA | MetUM 7.3 | Morgenstern et al. (2009, 2013); Stone et al. (2016) | UMUKCA-UCAM |
| CCSRNIES MIROC3.2 | 3.2 | Imai et al. (2013); Akiyoshi et al. (2016) | CCSRNIES |
| CESM1 CAM4-chem | CCMI_23 | Tilmes (2015b) | CAM3.5 |
| CESM1 WACCM | CCMI_30 | Solomon et al. (2015); Garcia et al. (2016); Marsh et al. (2013) | WACCM |
| CHASER (MIROC-ESM) | 4.5 | Sudo et al. (2002); Sudo and Akimoto (2007); Watanabe et al. (2011); Sekiya and Sudo (2012, 2014) | N/A |
| CMAM | v2.1 | Jonsson et al. (2004); Scinocca et al. (2008) | CMAM |
| CNRM-CM | v5-3 | Voldoire et al. (2012); Michou et al. (2011), http://www.cnrm-game-meteo.fr/spip.php?rubrique235 | CNRM-ACM |
| EMAC | v2.51 | Jöckel et al. (2010); Jöckel et al. (2016) | EMAC |
| GEOSCCM | 3 | Molod et al. (2012, 2015); Oman et al. (2011, 2013) | GEOSCCM |
| GFDL-AM3 | 3 | Donner et al. (2011); Lin et al. (2014, 2015a, b) | AMTRAC |
| GFDL-CM3 | 3 (CMIP5) | Griffies et al. (2011); John et al. (2012); Levy et al. (2013) | AMTRAC |
| HadGEM3-ES | HadGEM3 GA4.0, NEMO 3.4, CICE, UKCA, MetUMvn8.2 | Walters et al. (2014); Madec (2008); Hunke and Lipscombe (2008); Morgenstern et al. (2009); O'Connor et al. (2014); Hardiman et al. (2016) | UMUKCA-METO |
| LMDz-REPROBUS-CM5&CM6 | IPSL-CM5 & CM6 | Marchand et al. (2012); Szopa et al. (2012); Dufresne et al. (2013). No reference yet on CM6. | LMDz-REPROBUS |
| MRI-ESM1r1 | 1.1 | Yukimoto et al. (2012, 2011); Deushi and Shibata (2011) | MRI |
| MOCAGE | 2.15.1 | Josse et al. (2004); Guth et al. (2016) | N/A |
| SOCOL | 3 | Revell et al. (2015); Stenke et al. (2013) | SOCOL v.2 |
| TOMCAT | 1.8 | Chipperfield (1999, 2006) | N/A |
| ULAQ-CCM | 3, yr 2012 | Pitari et al. (2014) | ULAQ |
| UMSLIMCAT | 1 | Tian and Chipperfield (2005) | UMSLIMCAT |
| UMUKCA-UCAM | MetUM 7.3 | Morgenstern et al. (2009); Bednarz et al. (2016) | UMUKCA-UCAM |



**Table 3.** Governing equations, horizontal discretization, and vertical grid of the atmosphere component of models. NH = non-hydrostatic. PE = primitive equations. CTM = chemistry-transport model. QG = quasi-geostrophic. F[D,V]LL = finite [difference, volume] on lat-lon grid. STQ = spectral transform quadratic. STL = spectral transform linear.[f][g][h][i][j] CP = Charney-Phillips. TA = hybrid terrain-following altitude. TP = hybrid terrain-following pressure. NTP = non-terrain following pressure. FVCS = finite volume cubed sphere. T21 ≈ 5.6° × 5.6°. T42 ≈ 2.8° × 2.8°. T47 ≈ 2.5° × 2.5°. T63 ≈ 1.9° × 1.9°.

| Model name | Gov. eq. | Hor. disc. | Resolution | Vert. grid | Top level | Top of model | Coord. sys. | Comment |
|---|---|---|---|---|---|---|---|---|
| ACCESS CCM | | | | | | | | |
| NIWA-UKCA | NH | FDLL | 3.75° × 2.5° | CP60 | 84 km | 84 km | TA | Arakawa-C |
| UMUKCA-UCAM | | | | | | | | |
| CCSRNIES MIROC3.2 | PE | STQ | T42 | L34 | 1.2 Pa | 1 Pa | TP | |
| CHASER (MIROC-ESM) | PE | STQ | T42 | L57 | 56 km | 56 km | TP | Arakawa-C |
| CESM1 CAM4-chem | PE | FVLL | 1.9° × 2.5° | L26/56 | 200 Pa | 100 Pa | TP | Lin et al. (2004) |
| CESM1 WACCM | PE | FVLL | 1.9° × 2.5° | L66/88 | 140 km | 140 km | TP | Lin et al. (2004) |
| CMAM | PE | STL | T47 | L71 | 0.08 Pa | 0.0575 Pa | TP | |
| CNRM-CM5-3 | PE | STL | T63 | L60/89 | 7/8 Pa | 0 Pa | TP | |
| EMAC | PE | STQ | T42 | L47/90 | 1 Pa | 0 Pa | TP | |
| GEOSCCM | PE | FVCS | ∼ 2° × 2° | L72 | 1.5 Pa | 1 Pa | TP | |
| GFDL-AM3/CM3 | NH | FVCS | ∼ 2° × 2° | L48 | 86 km | 86 km | TA | Donner et al. (2011 |
| HadGEM3-ES | NH | FDLL | 1.875° × 1.25° | CP85 | 85km | 85km | TA | Arakawa-C |
| MRI-ESM1r1 | PE | STL | TL159 | L80 | 1 Pa | 0 Pa | TP | |
| LMDz-REPROBUS | PE | FVLL | 3.75° × 2.5° | L39/79 | ∼70/80 km | ∼70/80 km | TA | Arakawa-C |
| MOCAGE | CTM | FDLL | 2° × 2° | L47 | 500 Pa | 500 Pa | TP | |
| SOCOL | PE | STL | T42 | L39 | 1 Pa | 0 Pa | TP | |
| TOMCAT | CTM | FVLL | 2.8° × 2.8° | L60 | 10 Pa | 0 Pa | TP | ERA-Interim |
| ULAQ CCM | QG | STL | T21 | CP126 | 4 Pa | 4 Pa | NTP | |
| UMSLIMCAT | PE | FDLL | 3.75° × 2.5° | L64 | 1 Pa | 0.77 Pa | TP | Arakawa-B |





**Table 4.** Reference simulations, by model. Simulations in brackets are incomplete at the time of publication.

| Model name | REF-C1 (1960-2010) | REF-C2 (1960-2100) | REF-C1SD (1980-2010) |
|---|---|---|---|
| ACCESS CCM | 1 | 2 | |
| CCSRNIES MIROC3.2 | 3 | 1 | 1 |
| CESM1 CAM4-chem | 3 | 3 | 1 (NASA MERRA) |
| CESM1 WACCM | 5 | 3 | 1 (NASA MERRA) |
| CHASER (MIROC-ESM) | 1 | 1 | |
| CMAM | 3 | 1 | 1 |
| CNRM-CM5-3 | 4 | 2 | 2 |
| EMAC | 2 | 3 | 4 |
| GEOSCCM | 1 | 1 | 1 |
| GFDL-AM3 | | | 1 (Lin et al., 2014) |
| GFDL-CM3 | | 5 | |
| HadGEM3-ES | 1 | 1 (+2) | (2) |
| LMDz-REPROBUS | 1 (L39) | 1 (L39) | 1 (L39) |
| MRI-ESM1r1 | 1 | 1 | 1 |
| MOCAGE | 1 | (1) | 1 |
| NIWA-UKCA | 3 | 5 | |
| SOCOL | 1 | 1 | |
| TOMCAT | | | 1 |
| ULAQ CCM | 3 | 3 | |
| UMSLIMCAT | 1 | 1 | |
| UMUKCA-UCAM | 1 | 2 (+5 1980-2080) | 1 |
| total | 36 | 45 | 19 |

**Table 5.** SEN-C1 sensitivity simulations, by model.

| Model name | SEN-C1-Emis | SEN-C1SD-Emis | SEN-C1-fEmis | SEN-C1SD-fEmis | SEN-C1-SSI |
|---|---|---|---|---|---|
| CESM1 CAM4-chem | | | | 3 | |
| CHASER (MIROC-ESM) | | | | 1 | |
| GFDL-AM3 | 5 (Lin et al., 2014) | 1 | | 1 | |
| MOCAGE | | | | 1 | |
| NIWA-UKCA | | | 1 | | |
| TOMCAT | | | | | 1 |
| ULAQ CCM | | | 1 | | 3 |
| UMSLIMCAT | 1 | | 1 | | 1 |
| total | 6 | 1 | 3 | 6 | 5 |



**Table 6.** SEN-C2 sensitivity simulations, by model.

| Model name | RCP2.6 | RCP4.5 | RCP8.5 | fODS | fODS2000 | fGHG | fEmis | GeoMIP | SolarTrend | fCH4 | fN2O |
|---|---|---|---|---|---|---|---|---|---|---|---|
| ACCESS CCM | | | | 2 | | 1 | | | | | |
| CCSRNIES MIROC3.2 | 1 | 1 | 1 | 1 | 1 | 1 | | | 1 | | |
| CESM CAM4-chem | | | | | | | | 3 (1°) | | | |
| CESM1 WACCM | | 1 | 3 | 3 | 3 | 3 | | | | | |
| CHASER (MIROC-ESM) | 1 | 1 | 1 | 1 | | 1 | 1 | (1) | | | |
| CMAM | 1 | 1 | 1 | 1 | | 1 | | | | | |
| GFDL-CM3 | 1 | 3 | 1 | | | | | | | | |
| HadGEM3-ES | | | | | (3) | (3) | | | | | |
| LMDz-REPROBUS | 1(L39) | 1(L39) | 1(L39) | 1(L39) | 1(L39) | 1(L39) | | | | | |
| NIWA-UKCA | | | | 2 | | 3 | | | | (1) | (1) |
| SOCOL | 1 | 1 | 1 | | | | 1 | | | 1 | 1 |
| ULAQ CCM | 1 | 1 | 1 | 1 | 1 | 1 | 2 | 2 | 3 | | |
| total | 7 | 10 | 10 | 12 | 9 | 15 | 4 | 6 | 4 | **2** | **2** |

**Table 7.** Summary of forcings and emissions data used in the GFDL-AM3 hindcast simulations, with italics indicating where the data differ from the CCMI recommendations. L2014 = Lin et al. (2014).

| Experiment | L2014 name | Meterology | Period | RF | CH$_4$ | Anth. emiss. |
|---|---|---|---|---|---|---|
| REF-C1SD | BASE | NCEP $u$ & $v$ | 1980-2010 | REF-C1 | REF-C1 | REF-C1 (Except SO$_2$, BC, and OC after 1996) |
| SEN-C1SD-fEmis | FIXEMIS | as REF-C1SD | 1980-2010 | REF-C1 | *2000\** | *1970-2010 climatology\** |
| SEN-C1SD-Emis | IAVFIRE | as REF-C1SD | 1980-2010 | CCMI | *2000\** | *1970-2010 climatology\** |
| SEN-C1-Emis | AMIP | N/A | 1960-2010 | REF-C1 | *2000\** | *O$_3$ precursors: FIXEMIS; aerosol precursors: REF-C1* |