# Peer review of "Review of the global models used within the Chemistry-Climate Model Initiative (CCMI)"

_Geoscientific Model Development, 2016_

## Referee Comment (RC1) · Anonymous Referee #1 · 2 Nov 2016

Review of "Review of global mean models used within the Chemistry-Climate Model Iniative (CCMI)" by Morgenstern et al.

This manuscript provides information on the global chemistry models that have supplied simulation data for the CCMI phase 1 simulations (CCMI-1), as well as some detail on the set up of the various simulations, including emissions, sea surface temperature and sea ice boundary conditions, and other driving data. Papers of this sort can be an invaluable resource for scientists analyzing multi-model output, especially when they record important distinctions between the model as used for the study, compared to the version that might be described in the standard reference.

This manuscript does contain a lot of useful information on the different models used for CCMI-1, and the authors must be commended for what must have often been a

tedious project! However, I believe it could be improved if it was organized slightly differently, and with a more proportionate/balanced amount of space spent describing each model and process. I would also recommend that the authors read through the final text carefully, as it occasionally reads like a final draft rather than a final version. The manuscript is otherwise extremely useful and uncontroversial, and I would not foresee there being any issue with its eventual publication.

I have detailed my general criticisms below, together with a non-exhaustive list of line-by-line comments (many of which are catching specific instances of a general malady).

General criticisms 1. The time spent discussing the different aspects and different models seems a little disproportionate. For instance, there is often reference to the specific aspects of the MetUM family of models, but not the same information for the others. With regard to the detail on the forcings, the section on solar forcing provides a good amount of detail, whereas the information is more scant in some of the other sections (e.g., ozone precursor emissions). One suggestion here would be to describe the general case and list the models that differ from that, including in the way that they do. This is the kind of information that the reader will need to be able to get clearly and easily when they are considering their own analysis of the ensemble. (I appreciate that this is about balancing what to have in the supplement vs the main text.)

2. A suggestion related to the above: The authors could consider putting the description of the CCMI-1 simulations/scenarios first before launching in to describing the models in general (resolution etc.) and then how each one might be set up differently for a given boundary condition or forcing file. For this second part, the authors can then (for example) have a sub-section that describes the aerosol modules and aerosol emissions, saying how the latter are differently implemented depending on the inter-model differences in the former. This way the reader only has to look in one place to understand model differences and implementation differences for the given area of interest.

3. Description of the models. Not all the models are CCMs I believe (e.g. P1, L16), in that not all couple chemistry and climate throughout the whole atmosphere. In chapter 7 of the (unfinished) IGAC TOAR project, there is a suggestion for a naming convention for global chemistry models: a "CCM" is where chemistry is coupled to climate in a two-way matter (although is this the case throughout the atmosphere for all the models here?); a "chemistry GCM" or "GCM with chemistry" is where the chemical changes do not feed back onto the climate (e.g. UM-CAM); and a "CTM" is a different model framework that uses (elements of) offline meteorology to drive transport, temperatures for chemistry etc.

4. A general point about CCMI that the authors should get across: unlike CMIP5, CCMVal, ACCMIP, HTAP etc, CCMI is not a "MIP", or even a modeling experiment. Rather it is a community that addresses questions related to chemistry-climate issues, by running, analyzing, evaluating and improving global chemistry models (cf. GEIA and iLEAPS for example). As such, the present manuscript should be clear that it is describing the models and experiment set up for the CCMI Phase 1 simulations (CCMI-1). It could also make reference to the phase 2 experiment, being conducted jointly with Aercom and representing the chemistry and aerosol communities' contribution to CMIP6 (AerChemMIP). Phase 2/AerChemMIP might involve different models, so we should perhaps be careful not to talk about "CCMI models".

5. Not a serious criticism for such a paper, but there are often statements made without citation or justification. Some examples are given below, but could the authors please check this when editing (e.g., the importance of clouds for climate sensitivity etc.)

Specific comments and technical corrections P1, L5: not just air quality, but also tropospheric composition (also in the Introduction)

P2, L11: "Previous generation models" – when? Reference?

P2, L12: Suggest rephrasing the "multi-faceted" bit

**GMDD**

P2, L22: Delete "activities"

P2, L25 (and throughout): I understand that it is convenient to refer to CCMVal-2 for previous model descriptions, but not all the CCMI-1 models took part in that experiment. This is acknowledged in some places, but perhaps just better to remove some of the references to CCMVal-2 as this is different.

P2, L32: The paper -> This paper (or "The present paper"?)

P3, L6: MOCAGE may not have been in CCMVal-2, but it was in the other CCMI-1 forerunner, ACCMIP.

P3, L20: "impact of ozone..." – Clearer/more general example perhaps?

P4, L5: improved -> increased

P4, L6: what was the baseline for these resolution "improvements"?

P4, L10: where do these exception models stop? (Also "cf." -> "see"? Since cf. means compare with)

P4, L15: SD runs – are all the models nudged, or are some actually closer to CTMs? (CESM family perhaps?)

P4, L15: QBO "may not" require specific forcing – don't we know this for certain? (There are other examples of the use of "may", when these should be nailed down!)

P4, L29: How similar are the "MetUM" family of models? Could the authors comment (and include in the text) on how many other families there are? What are the implications for model independence and assessing structural uncertainty in the CCMI-1 ensemble? ...In any case, could these "families" be identified up front to help the non-expert reader?

P5, L27: Unified Model based -> MetUM (?) ...Another example of MetUM being singled out, with only a brief mention of 2 other model families (SOCOL and CESM).

P6, L5: Spell out "w.r.t." (other occasions too)

P6, L12: "constituent fields" – which constituents are meant?

P6, L15: How are the emissions done for the lumped mechanisms? Do models just emit the reactive carbon for the species that they have, or is the total amount proportionally shared out over the available VOCs? With reference to my earlier general comment, this could be a good example of where a description of a process could be combined with a description of the specifications for the simulations.

P6, L22: Are CH4 *concentrations* prescribed?

P7, L2: "models vary" – how?

P7, L13: Consider (briefly) describing what is meant by "numerical diffusion"

P8, L10: Reference for statement about clouds and climate sensitivity?

P9, L1 and L6: What about SW and LW treatments for non-SPARC models?

P10, L3: Reference for NEMO?

P10, L8: Do the models apply the solar forcing to photolysis? (I know this is in Table S29, but good to flag here)

P11, L15: similar -> the same as

P11, L17: More specifics for the solarTrend simulation?

P11, L21: Helpful if spelt out what this simulation is

P13, L8 and L11: "essentially" is imprecise

P13, L10: Consider a map of NOx emissions changes over different time periods, since it shows the complexity of the changes for anthropogenic emission changes. Similarly for this section and the following one, the authors could discuss how the differences in the model chemistry schemes can result in a large inter-model differences in the actual

VOC emission flux (NOx and CO to a lesser extent).

P13, L17: To be clear: is it that REF-C2 uses MACCity to 2000 and then RCP6.0, and REF-C1 uses MACCity to 2010?

P13, L26: Reference for MEGAN?

P14, L3: "is recommended" – what else was used, and by which model/what models?

P14, Fig 2: Suggest that contours are overlaid as lines, else the non-significant trends and those between +/-0.1 are not separated. Masking of non-significant trends might be better done with (say) a gray palette with this color bar.

P16, L8 and L9: "can be" and "should be", but what was actually done?

P17, L9 and L10: has -> have (verb agreement with "models" not "a majority"/"a subset")

P17, L14: "Some models" – which?

P18, L7: "experience" by who? Ref? Folklore?

P18, L16: are advised -> should

P18, Appendix A: The model information varies wildly in detail. Can this be harmonized? Erring on more detail (e.g. CESM) would be useful.

P25, Appendix B: Will CCMI be keeping a website with a list of known issues that arise after the publication of this description?

Tables (including Supplement): Could the authors please review the captions to ensure that they can be understood without too much cross-referencing. Table S5 seems particularly esoteric, including the abbreviations in the table itself (FFSL?). In addition the citations are often missing brackets etc. Perhaps not a big deal you might think, but this is a subject that is about attention to detail and it gives the reader confidence that the information has been compiled with care!

---

## Referee Comment (RC2) · Anonymous Referee #2 · 12 Nov 2016

This paper presents a technical overview of the models and simulations that contribute to the Chemistry-Climate Model Initiative (CCMI). In particular, the authors highlight changes and improvements to the models since CCMVal-2.

This paper contains a vast array of useful information that I anticipate will be widely used as a point of reference by many in the chemistry-climate modeling community. I recommend publication after my comments below have been addressed.

**General comments**
1. Firstly, I suggest some structural changes - mainly in Section 3 and related Tables – to enhance readability. A few points:

(a) Why is Section 2 standalone? Why not use this as an introduction in Section 3, (since both sections describe model details)? Then, the paper would be neatly partitioned into sections describing model details (Sect. 2), simulations (Sect. 3) and forcings (Sect. 4).

(b) I do not find the ordering sensible of Sect. 3 sensible. It is currently difficult for a reader to quickly obtain information about any particular aspect of the models. How about ordering such that similar subsections appear close together, especially since there are so many of them. For example, consider the ordering below (but feel free to make changes around this general idea):
-Grid and numerical methods: general model makeup, model resolution, advection, timestepping and calendars, horizontal diffusion
-Dynamical and physical processes: QBO, gravity wave drag, physical parameterizations, cloud microphysics
-Chemistry, aerosols, radiation: trop chemistry, strat chemistry, heterogeneous chemistry, PSCs, trop aerosols, volcanic effects, photolysis, SW, LW, solar forcing
-Coupling / other boundary conditions: ocean surface / ocean coupling, land surface

(c) The tables in the Supplement should more or less follow the above order. Consider also adding an explicit sentence at the beginning of each subsection that references the appropriate supplemental table e.g. 'Table Sx shows …'.

(d) Similarly, in Section 4, I would re-order such that the simulations in which one set of boundary conditions is kept fixed (fGHG, fODS, fEMIS, fCH4, fN2O) appear together, as do the simulations in which time-varying perturbations are applied (C1-Emis, C2-RCP, C2-GeoMIP, C1-SSI, C2-SolarTrend).

2. My second general point is to include more detail throughout the paper. This includes, but is not limited to, my questions in the Specific comments below. This also applies to Table captions (including expansion of abbreviations); the reader cannot be expected to reference the CCMVal-2 report. Finally, this applies to Appendix A e.g. no details on the chemical species or mechanisms is not provided for most models; I understand that this is might be a tough task given the large number of different modeling groups involved. On that note, I do commend the authors on well documenting the changes in each model since CCMVal-2.

**Specific comments**
I organize the following points primarily by section, and page/line number where necessary:
(3.1) General model make-up:
-I'm not entirely sure about the point of this subsection. Much of it could be neatly partitioned elsewhere. Perhaps just include some general comments on the components/coupling in the models and retain the text on familial relationships.
-First few lines are repeated in Section 3.12 (Ocean Forcing). Could remove details here and reference that section.
-P3 L20: 'the impact of ozone depletion on surface climate is represented consistently': a bit unclear. Consider making a broader statement (first): 'surface

climate is able to respond to changes in atmospheric composition, some of which may be considered climate feedbacks', or simply make the point that climate feedbacks are self-consistently incorporated.
-P3 L28: would make sense to include grid details with model resolution (Sect. 3.2).

(3.2) Model resolution:
P4 L9: from Table 3, it looks like CNRM-CM5-3 and TOMCAT also do not completely cover the stratosphere (I'm defining a stratopause at ~50km, 1hPa).

(3.3) QBO:
-P4 L15: '...which means the QBO may not require explicit forcing to occur in the models, or it may be absent': I don't understand this sentence. Please clarify.
-Mention which models do not have a QBO at all and which (few!) internally generate a QBO either in the text or tables.

(3.4) Volcanic effects:
-Clean up slightly: highlight, in turn, which (or how many) models include (a) online volcanic aerosols, (b) impose offline aerosols (heating rates) and (c) do not have any representation of radiative effects from volcanic aerosols. This will correspond much better with Table S4.

(3.5) Advection:
-Expand on the 'different settings for hydrological and chemical tracers' in the MetUM models or add reference.

(3.8) Tropospheric aerosols:
-Combine this section with Sect. 3.21.
-Would it be worth mentioning the main tropospheric aerosol species / heterogeneous reactions included in the models?

(3.9) Stratospheric chemistry
-P5 L28: 'to lump all': is it really all, or most?
-Reference for how Cl source gases are lumped?
-Are Br source gases also lumped in some cases?
-P5 L30: reference for the recommendation for Br species?
-Besides halogen chemistry, can you briefly describe the differences/commonalities in stratospheric chemistry between the models? E.g. how is $CH_4$ is oxidized to stratospheric water vapor in the models?
-How is stratospheric chemistry represented for the models that do not cover the full vertical extent of the stratosphere?

(3.11) Strat/trop heterogeneous chemistry
-Provide more details of $SO_2$ -> $SO_3$ oxidation (e.g. is it with interactive or offline oxidant fields?).

(3.15) Cloud microphysics
-P8 L10: Add reference(s) to first sentence.

(3.15) and (3.16)
-Little detail provided on cloud and land surface schemes – elaborate if practical.

(3.17) PSCs
-For the non-expert, elaborate on what is calculated assuming 'thermodynamic equilibrium'. Can mention formation of NAT/ice PSCs, and how these differ between models.
-It would make sense for this section to be near Sect. 3.11 (heterogeneous chemistry)

(3.22) Ocean coupling
-This section should be combined with Sect. 3.12 (ocean surface forcing) or appear close to it.
-Mention also the sea ice modules / boundary conditions here.

(3.23) Solar forcing
-In cases where SW radiation and photolysis are not handled consistently, what are the radiation schemes for photolysis? For these photolysis schemes, can we assume the effects of the 11-yr solar cycle are not included?

(4) CCMI simulations
-P10 L15: 'Forcings are discussed briefly...' -> 'The specific forcings imposed are discussed briefly...'.
-P10 L18-20: This sentence is very unclear and should probably be separated into at least 3 sentences. Do you mean that ODS concentrations (or EESC), rather than emissions, peak around yr 2000? Which 'industrial emissions' do you mean? Separate the discussion of GHGs from ODSs and the other 'emissions' that are referred to. Should this sentence be in the Forcings section?
-P10 L22: clarify that SD stands for "specified dynamics".
-P10 L22: differences in dynamics between nudged and free-running models are not necessarily due to inherent dynamical biases in the model; they could also be due to the greater presence of internal variability in the free-running case.
-P11 L3: which emissions?
-P11 L5: I'm confused as to the exact forcings imposed here: GHGs? SSTs and sea ice for models not coupled to an ocean? ODSs (including those that are not GHGs)? NOx? NMHCs?
-P11 L9: 'sea surface' -> 'sea surface and sea ice'.
-P11 L11: only surface emissions, or also 3D emissions (e.g. aircraft NOx)?
-P11 L15: clarify that SSI stands for Spectral Solar Irradiance.

(5) Forcings
-P12 L1-2: clarify that N2O and CH4 boundary conditions refer to surface mass mixing ratios.
-P13 L9: add reference.
-P13 L14: 'cause NOx emissions to peak and then decline' - clarify that this is only an assumption for the future.
-P13 L26: reference for MEGAN model
-P13 L19-30: which dataset(s) is/are used for historic emissions in which biogenic emissions are not interactively computed?
-P14 L10: reference appropriate table
-P15 L1-15: much of this information is provided in Sect. 3.21. Instead of repetition, talk here about the time series of aerosol precursor emissions (e.g. the projected reduction in future aerosol emissions over certain regions as with ozone precursors).

Tables
-Table 2: Is it possible to list (alongside the model names) the versions used for CCMVal-2, and, where relevant, the name of the ESM? Right now, there is little consistency.
-Table 4: caption: state that the numbers in the table stand for (I'm guessing) the number of ensemble members.
-Table 5 and 6: why are some numbers in bold and in brackets? What does L39 stands for?
-Table S2 caption: clarify that the numbers represent number of grid boxes.
-Table S5 caption: expand on abbreviations used in the table e.g. SL = semi-lagrangian etc.
-Table S5: for CESM models, CAM4 describes the atmospheric component but not the transport scheme (is it SL? please check.)
-Table S26: would make sense to keep this next to Table S5.

All tables: please try to find more references for each model (e.g. for the aerosol schemes in Table S4).

Appendices
-P22 L22-24: The two first sentences are already mentioned in the previous paragraph.
-P23 L15: citation for Wesely scheme.
-P25 L29: why are two of the simulations in bold?
-P26 L24: elaborate.

**Technical comments**
-P4 L12: misspelled 'oscillation'
-P5 L18: 'included' -> 'include'
-P5 L27: 'Unified Model' -> 'MetUM' for consistency
-P9 L2: Remove comma after 'table S17'
-P10 L8: 'increasing' -> increasingly'
-P19 L13: "Earth'" -> "Earth's"
-P24 L18: 'CTM be' -> 'CTM can be'
-Table S3 caption: 'CCM name' -> 'model name'

---

## Author Comment (AC1) · 16 Dec 2016

We thank the referee for his/her detailed and thoughtful comments.

*This manuscript provides information on the global chemistry models that have supplied simulation data for the CCMI phase 1 simulations (CCMI-1), as well as some detail on the set up of the various simulations, including emissions, sea surface temperature, sea ice boundary conditions, and other driving data. Papers of this sort can be an invaluable resource for scientists analyzing multi-model output, especially when they record important distinctions between the model as used for the study, compared to the version that might be described in the standard reference.*

*This manuscript does contain a lot of useful information on the different models used for CCMI-1, and the authors must be commended for what must have often been a*

*tedious project! However, I believe it could be improved if it was organized slightly differently, and with a more proportionate/balanced amount of space spent describing each model and process. I would also recommend that the authors read through the final text carefully, as it occasionally reads like a final draft rather than a final version. The manuscript is otherwise extremely useful and uncontroversial, and I would not foresee there being any issue with its eventual publication.*

We have reorganized the text slightly; in particular, we have reordered the paragraphs (also in response to reviewer 2's comment). We have proof-read the manuscript again.

*I have detailed my general criticisms below, together with a non-exhaustive list of line-by-line comments (many of which are catching specific instances of a general malady).*

*General criticisms*

1. *The time spent discussing the different aspects and different models seems a little disproportionate. For instance, there is often reference to the specific aspects of the MetUM family of models, but not the same information for the others. With regard to the detail on the forcings, the section on solar forcing provides a good amount of detail, whereas the information is more scant in some of the other sections (e.g., ozone precursor emissions). One suggestion here would be to describe the general case and list the models that differ from that, including in the way that they do. This is the kind of information that the reader will need to be able to get clearly and easily when they are considering their own analysis of the ensemble. (I appreciate that this is about balancing what to have in the supplement vs the main text.)*

   Indeed it has been our philosophy to report on models by exception, i.e. to discuss the general case and in addition only single out those models that deviate from that. We have reassessed the whole text with a view towards adding more detail. As for the section on emissions, this has already been detailed by Eyring et al. (2012); hence the discussion is kept short here. It has been our ambition to

give equal treatment to all models; hopefully the various minor changes applied to the text have levelled out any perceived inequalities.

2. *A suggestion related to the above: The authors could consider putting the description of the CCMI-1 simulations/scenarios first before launching in to describing the models in general (resolution etc.) and then how each one might be set up differently for a given boundary condition or forcing file. For this second part, the authors can then (for example) have a sub-section that describes the aerosol modules and aerosol emissions, saying how the latter are differently implemented depending on the intermodal differences in the former. This way the reader only has to look in one place to understand model differences and implementation differences for the given area of interest.*

The organization generally follows Morgenstern et al. (2011) but of course this is a matter of taste. The majority of model aspects discussed here is not associated with any external forcings. We accept that for some topics the reader has to look in two different places to find relevant information (one for how a particular aspect is implemented in the models and one for information on scenarios and forcings). The forcings generally define a storyline (such as best-of-knowledge reproduction of the past, middle-of-the-road anthropogenic-only climate forcings, hypothetical scenarios such as fixed-GHGs, etc). We feel it is best to leave these things separate from model definitions.

3. *Description of the models. Not all the models are CCMs I believe (e.g. P1, L16), in that not all couple chemistry and climate throughout the whole atmosphere. In chapter 7 of the (unfinished) IGAC TOAR project, there is a suggestion for a naming convention for global chemistry models: a "CCM" is where chemistry is coupled to climate in a two-way matter (although is this the case throughout the atmosphere for all the models here?); a "chemistry GCM" or "GCM with chemistry" is where the chemical changes do not feed back onto the climate (e.g.*

[Figure]

*UM-CAM); and a "CTM" is a different model framework that uses (elements of) offline meteorology to drive transport, temperatures for chemistry etc.*

Thanks for alerting us to this more precise nomenclature. Indeed for some models, composition is prescribed throughout parts of the domain (e.g. LMDZ-REPROBUS). We still refer to this model as a "CCM". There is no "UM-CAM"-type model here, i.e. all "CCMs" interactively feed ozone and other GHGs into radiation. We believe our usage of the terms "CTM" and "CCM" is consistent with the TOAR definition.

4. *A general point about CCMI that the authors should get across: Unlike CMIP5, CCMVal, ACCMIP, HTAP etc, CCMI is not a "MIP", or even a modeling experiment. Rather it is a community that addresses questions related to chemistry-climate issues, by running, analyzing, evaluating and improving global chemistry models (cf. GEIA and iLEAPS for example). As such, the present manuscript should be clear that it is describing the models and experiment set up for the CCMI Phase 1 simulations (CCMI-1). It could also make reference to the phase 2 experiment, being conducted jointly with Aercom and representing the chemistry and aerosol communities' contribution to CMIP6 (AerChemMIP). Phase 2/AerChemMIP might involve different models, so we should perhaps be careful not to talk about "CCMI models".*

We now use the term "CCMI-1" in the sense suggested by the reviewer where we feel that a distinction needs to be made between CCMI-1 and CCMI-2. We have also added a sentence on CCMI-2 in the introduction.

5. *Not a serious criticism for such a paper, but there are often statements made without citation or justification. Some examples are given below, but could the authors please check this when editing (e.g., the importance of clouds for climate sensitivity etc.)*

Upon re-reading the paper, we have tried to address this concern.

Specific comments and technical corrections

*P1, L5: not just air quality, but also tropospheric composition (also in the Introduction)*

We have added "tropospheric composition" which is more general than air quality.

*P2, L11: "Previous generation models" – when? Reference?*

We have added references to Morgenstern et al. (2010) (the CCMVal-2 model description paper) and Lamarque et al. (2013) – the ACCMIP reference.

*P2, L12: Suggest rephrasing the "multi-faceted" bit*

We have cut out this phrase – it seems unnecessary.

*P2, L22: Delete "activities"*

Done.

*P2, L25 (and throughout): I understand that it is convenient to refer to CCMVal-2 for previous model descriptions, but not all the CCMI-1 models took part in that experiment. This is acknowledged in some places, but perhaps just better to remove some of the references to CCMVal-2 as this is different.*

We accept that CCMVal-2 is significantly different from CCMI, and we have trimmed the references to CCMVal-2 to where it makes sense (keeping in mind not all models participated in that). However, there is only one CCM (CHASER) and two CTMs which had not participated in CCMVal-2. The remaining 17 models had all participated, and had been well characterized in their CCMVal2 versions. For this reason the whole paper was designed as a "sequel" to Morgenstern et al. (2010). We feel it does make sense to retain most references to CCMVal-2, particularly as in some respects the models have not significantly changed since CCMVal-2. A comparison to CCMVal-2 is also useful to characterize community-wide progress in the intervening 6 years.

*P2, L32: The paper → This paper (or "The present paper"?)*

This refers to Morgenstern et al. (2010) so not the "present" paper.

*P3, L6: MOCAGE may not have been in CCMVal-2, but it was in the other CCMI-1 forerunner, ACCMIP.*

This is now spelled out explicitly.

*P3, L20: "impact of ozone" – Clearer/more general example perhaps?*

This line got removed, in response to reviewer 2's comment.

*P4, L5: improved → increased*

Done.

*P4, L6: what was the baseline for these resolution "improvements"?*

In most cases the improvements refer to the CCMVal-2 versions. In the case of MOCAGE, relative to the ACCMIP version there is no improvement. This is now made explicit.

*P4, L10: where do these exception models stop? (Also "cf." → "see"? Since cf. means compare with)*

We now explicitly state the model top pressures for these two models.

*P4, L15: SD runs – are all the models nudged, or are some actually closer to CTMs? (CESM family perhaps?)*

The details of nudging vary, as is spelled out in the text. Indeed two of the models are CTMs (which one could think of as an extreme case of nudging). These models only completed the specified-dynamics simulations, which this section on QBO forcing does not apply to. None of the remaining models can be considered CTMs as the QBO nudging is only used in a portion of the model domain.

*P4, L15: QBO "may not" require specific forcing – don't we know this for certain? (There are other examples of the use of "may", when these should be nailed down!)*

We have removed this occurrence of "may". There are two other occurrences of "may" in the text which we consider are both justified.

*P4, L29: How similar are the "MetUM" family of models? Could the authors comment (and include in the text) on how many other families there are? What are the implications for model independence and assessing structural uncertainty in the CCMI-1 ensemble? In any case, could these "families" be identified up front to help the non-expert reader?*

The Unified Model family are all built around the Unified Model, but this model can be configured in various ways and is subject to ongoing development. So none of the members of this family are quite the same as others, but they do have considerable similarities. In the case of ACCESS CCM and NIWA-UKCA, the only difference is that ACCESS CCM is never coupled to an ocean/sea ice model, so these models for REF-C1 and SEN-C1 type simulations can be considered identical. They are therefore usually listed together. Then there is the CESM1 family (identified by name and usually listed together). Finally the ECHAM family – EMAC and SOCOL share a similar background climate model but use different chemistry packages. We now have a subsection specifically to document these similarities. Where the model formulations are identical, they are usually listed together in the tables, to aid the reader in understanding these similarities.

*P5, L27: Unified Model based → MetUM (?) Another example of MetUM being singled out, with only a brief mention of 2 other model families (SOCOL and CESM).*

We have strived to give all models equal treatment. The sentences are generally based on information provided by the model PIs. This paragraph got reformulated – only two models have increased the number of represented ODSs before participating in SPARC (2013). The other 4 already had a comprehensive list of species for CCMVal-

2. (We now use "MetUM" more consistently than before).

*P6, L5: Spell out "w.r.t." (other occasions too)*

We have expanded the two occurrences.

*P6, L12: "constituent fields" – which constituents are meant?*

The sentence has been rephrased. LMDz-REPROBUS does not handle tropospheric chemistry (below 400 hPa) at all, so all chemical fields are imposed there.

*P6, L15: How are the emissions done for the lumped mechanisms? Do models just emit the reactive carbon for the species that they have, or is the total amount proportionally shared out over the available VOCs? With reference to my earlier general comment, this could be a good example of where a description of a process could be combined with a description of the specifications for the simulations.*

There is no consistency regarding how this is handled, and there also was no recommendation regarding the details of this. Partly this is because the large diversity of tropospheric chemistry mechanisms in use for CCMI-1 makes such a standardization difficult to achieve. Eyring et al. (2013a) give a detailed account of how the emissions were defined.

*P6, L22: Are CH4 \*concentrations\* prescribed?*

No, mixing ratios are, in all but one case (CHASER). This has been corrected.

*P7, L2: "models vary" – how?*

This sentence has been rephrased. More detail has been added regarding the $SO_2 \rightarrow SO_3$ conversion process.

*P7, L13: Consider (briefly) describing what is meant by "numerical diffusion"*

We have added two sentences on "numerical diffusion".

*P8, L10: Reference for statement about clouds and climate sensitivity?*

We have added a reference here.

*P9, L1 and L6: What about SW and LW treatments for non-SPARC models?*

The CTMs do not explicitly consider this process. CHASER did not participate in either of the two identified precursor activities, so therefore progress cannot be quantified in the same manner as for the other models.

*P10, L3: Reference for NEMO?*

Added, also earlier for the first reference to NEMO.

*P10, L8: Do the models apply the solar forcing to photolysis? (I know this is in Table S29, but good to flag here) No, not generally, except where this is explicitly stated. We now explicitly name the models that treat photolysis and short-wave radiation consistently. This does not imply the use the same scheme for both; it may only imply that they use the same solar irradiance in both.*

*P11, L15: similar → the same as*

Changed.

*P11, L17: More specifics for the solarTrend simulation?*

We have added two more sentences and a reference to the SOLARIS website.

*P11, L21: Helpful if spelt out what this simulation is*

We have rephrased the sentence.

*P13, L8 and L11: "essentially" is imprecise*

We have removed both occurrences.

*P13, L10: Consider a map of NOx emissions changes over different time periods, since it shows the complexity of the changes for anthropogenic emission changes. Similarly for this section and the following one, the authors could discuss how the differences in*

*the model chemistry schemes can result in a large inter-model differences in the actual VOC emission flux (NOx and CO to a lesser extent).*

We have now included a new multi-panel plot showing annual-mean emissions of NOx and SO2 in different years spanning the REF-C2 period. As for the influence of chemistry on the actual emission flux, we agree that this is a major determinant of model behaviour. This is discussed to some extent in the subsection on tropospheric chemistry.

*P13, L17: To be clear: is it that REF-C2 uses MACCity to 2000 and then RCP6.0, and REF-C1 uses MACCity to 2010?*

Yes, that is correct. We have rephrased this sentence to make this explicit.

*P13, L26: Reference for MEGAN?*

Added (also in response to reviewer 2).

*P14, L3: "is recommended" – what else was used, and by which model/what models?*

Only LMDz-REPROBUS used AMIP. This is now spelt out in the subsection on surface forcings. Here we additionally now include a reference to this subsection.

*P14, Fig 2: Suggest that contours are overlaid as lines, else the non-significant trends and those between +/-0.1 are not separated. Masking of non-significant trends might be better done with (say) a gray palette with this color bar.*

It was actually intentional not to separate insignificant from small trends (so that all trends that appear in red or blue hues are significant). We now stipple the parts of the globe where SST trends are insignificant in the HadISST dataset, and have rephrased the caption.

*P16, L8 and L9: "can be" and "should be", but what was actually done?* We have rephrased these sentences.

*P17, L9 and L10: has → have (verb agreement with "models" not "a majority"/"a subset")* Done.

*P17, L14: "Some models" – which?*

We now spell out which models are meant here.

*P18, L7: "experience" by who? Ref? Folklore?*

We now provide a recent exemplary reference for this.

*P18, L16: are advised → should*

Done.

*P18, Appendix A: The model information varies wildly in detail. Can this be harmonized? Erring on more detail (e.g. CESM) would be useful.*

We have rephrased some minor aspects of these model descriptions, but in most cases PIs felt that their models were adequately described.

*P25, Appendix B: Will CCMI be keeping a website with a list of known issues that arise after the publication of this description?*

Good idea, we now give a link to a yet-to-be-populated website that lists such errors.

*Tables (including Supplement): Could the authors please review the captions to ensure that they can be understood without too much cross-referencing. Table S5 seems particularly esoteric, including the abbreviations in the table itself (FFSL?). In addition the citations are often missing brackets etc. Perhaps not a big deal you might think, but this is a subject that is about attention to detail and it gives the reader confidence that the information has been compiled with care!*

We have revised the tables and captions in light of this comment. We have provided explanations for numerous acronyms and names of some species. Some remain or are difficult to expand to a sensible meaning. For example, some species names of lumped

chemical compounds only make sense in the context of the corresponding chemical mechanism. In these cases, the readers are referred to the specialist literature. We have standardized the citation format.

---

## Author Comment (AC2) · 16 Dec 2016

We thank the review for his/her very constructive and useful comments.

*This paper presents a technical overview of the models and simulations that contribute to the Chemistry-Climate Model Initiative (CCMI). In particular, the authors highlight changes and improvements to the models since CCMVal-2. This paper contains a vast array of useful information that I anticipate will be widely used as a point of reference by many in the chemistry-climate modelling community. I recommend publication after my comments below have been addressed.*

*General comments*

1. *Firstly, I suggest some structural changes - mainly in Section 3 and related Tables*

[Figure]

*– to enhance readability. A few points:*

(a) *Why is Section 2 standalone? Why not use this as an introduction in Section 3, (since both sections describe model details)? Then, the paper would be neatly partitioned into sections describing model details (Sect. 2), simulations (Sect. 3) and forcings (Sect. 4).*

We have merged sections 2 and 3, to improve readability.

(b) *I do not find the ordering sensible of Sect. 3 sensible. It is currently difficult for a reader to quickly obtain information about any particular aspect of the models. How about ordering such that similar subsections appear close together, especially since there are so many of them. For example, consider the ordering below (but feel free to make changes around this general idea):*

- *Grid and numerical methods: general model makeup, model resolution, advection, time-stepping and calendars, horizontal diffusion*
- *Dynamical and physical processes: QBO, gravity wave drag, physical parameterizations, cloud microphysics*
- *Chemistry, aerosols, radiation: trop chemistry, strat chemistry, heterogeneous chemistry, PSCs, trop aerosols, volcanic effects, photolysis, SW, LW, solar forcing*
- *Coupling / other boundary conditions: ocean surface / ocean coupling, land surface*

We have reordered the subsections following the reviewer's suggestions.

(c) *The tables in the Supplement should more or less follow the above order. Consider also adding an explicit sentence at the beginning of each subsection that references the appropriate supplemental table e.g. 'Table Sx shows . . . '.*

We have revised the ordering of the tables and have improved the referencing. Generally, though, we find that the references linking the subsections to the supplementary tables are clear enough.

(d) *Similarly, in Section 4, I would re-order such that the simulations in which one set of boundary conditions is kept fixed (fGHG, fODS, fEMIS, fCH4, fN2O) appear together, as do the simulations in which time-varying perturbations are applied (C1-Emis, C2-RCP, C2-GeoMIP, C1-SSI, C2-SolarTrend).*

We follow the reviewer's suggestion.

2. *My second general point is to include more detail throughout the paper. This includes, but is not limited to, my questions in the Specific comments below. This also applies to Table captions (including expansion of abbreviations); the reader cannot be expected to reference the CCMVal-2 report. Finally, this applies to Appendix A e.g. no details on the chemical species or mechanisms is not provided for most models; I understand that this is might be a tough task given the large number of different modeling groups involved. On that note, I do commend the authors on well documenting the changes in each model since CCMVal-2.*

Indeed, we consider it impractical to document here full details of the chemical mechanisms used as these tend to be big, and there are a lot of models listed here. We feel this should be the subject of a separate paper. We have revised text in various places to provide more detail and improve clarity.

*Specific comments:*

*I organize the following points primarily by section, and page/line number where necessary:*

*(3.1) General model make-up:*

- *I'm not entirely sure about the point of this subsection. Much of it could be neatly partitioned elsewhere. Perhaps just include some general comments on the components/coupling in the models and retain the text on familial relationships.*

We have reorganized this into two newly titled subsections on family relationships and on the grids in use by the CCMI-1 models. The sentence on coupling to ocean models duplicates another sentence in the corresponding ocean coupling subsection.

- *First few lines are repeated in Section 3.12 (Ocean Forcing). Could remove details here and reference that section.*

  We have removed the duplication.

- *P3 L20: 'the impact of ozone depletion on surface climate is represented consistently': a bit unclear. Consider making a broader statement (first): 'surface climate is able to respond to changes in atmospheric composition, some of which may be considered climate feedbacks', or simply make the point that climate feedbacks are self-consistently incorporated.*

  We have rephrased this sentence.

- *P3 L28: would make sense to include grid details with model resolution (Sect. 3.2).*

  We have merged these two paragraphs into one subsection ('Atmosphere grids and resolution').

- *(3.2) Model resolution:*

- *P4 L9: from Table 3, it looks like CNRM-CM5-3 and TOMCAT also do not completely cover the stratosphere (I'm defining a stratopause at 50km, 1hPa).*

  These two models extend into the mesosphere (with tops at 7/8 Pa and 10 Pa respectively). We now consistently use units of Pa here, to avoid this confusion.

- *(3.3) QBO:*

- *P4 L15: '...which means the QBO may not require explicit forcing to occur in the models, or it may be absent': I don't understand this sentence. Please clarify.*

  In these models, the QBO is not externally imposed. This means it's either spontaneously occurring (with a degree of realism not assessed here) or it is absent. We have rephrased the sentence.

- *Mention which models do not have a QBO at all and which (few!) internally generate a QBO either in the text or tables.*

  We stated at the start of the paper that model evaluations (such as of the occurrence and realism of the QBO) would be out-of-scope for this paper. An assessment of which models spontaneously produce a QBO would then lead to a discussion of the quality of this simulation; we prefer to leave this to other CCMI papers to explore.

- *(3.4) Volcanic effects:*

- *Clean up slightly: highlight, in turn, which (or how many) models include (a) online volcanic aerosols, (b) impose offline aerosols (heating rates) and (c) do not have any representation of radiative effects from volcanic aerosols. This will correspond much better with Table S4.*

  We have revised this section.

- *(3.5) Advection:*

- *Expand on the 'different settings for hydrological and chemical tracers' in the MetUM models or add reference.*

  We have removed this sentence. Moisture and chemical tracers are advected consistently in these models. These models are based on a semi-Lagrangian dynamical core, and physical tracers (momentum, heat) are transported using different settings from the chemical and hydrological tracers. But this is not the

topic of this paragraph, and in other models these physical quantities would likely also be advected using schemes that differ from the tracer advection scheme.

- *(3.8) Tropospheric aerosols:*

- *Combine this section with Sect. 3.21.*

  Done.

- *Would it be worth mentioning the main tropospheric aerosol species / heterogeneous reactions included in the models?*

  We have added a sentence summarizing what's listed in table S18.

- *(3.9) Stratospheric chemistry*

- *P5 L28: 'to lump all': is it really all, or most?*

  We have cut out the word "all". The lumping makes sure that realistic amounts of chlorine and bromine enter the stratosphere.

- *Reference for how Cl source gases are lumped?*

  This is detailed in Morgenstern et al. (2009). Reference added.

- *Are Br source gases also lumped in some cases?*

  Yes they are. We have added a sentence to this effect.

- *P5 L30: reference for the recommendation for Br species?*

  We have added a reference for this (Eyring et al., 2013).

- *Besides halogen chemistry, can you briefly describe the differences/commonalities in stratospheric chemistry between the models? E.g. how is CH4 is oxidized to stratospheric water vapor in the models?*

The models generally employ a detailed methane oxidation scheme that represents intermediates of methane oxidation such as methyl hydroperoxide and formaldehyde. A sentence is added to this effect.

- *How is stratospheric chemistry represented for the models that do not cover the full vertical extent of the stratosphere?*

  With the exception of the upper-atmosphere chemistry of WACCM, there is no characteristic of the chemistry schemes discussed here that could be attributed to the lower top in the low-top models.

- *(3.11) Strat/trop heterogeneous chemistry*

- *Provide more details of SO2 → SO3 oxidation (e.g. is it with interactive or offline oxidant fields?).*

  We have added a few sentence on this process. In all cases where we now give details on this process, oxidants are calculated interactively.

- *(3.15) Cloud microphysics*

- *P8 L10: Add reference(s) to first sentence.*

  Reference added (IPCC 2013).

- *(3.15) and (3.16)*

- *Little detail provided on cloud and land surface schemes – elaborate if practical.*

  We feel that a discussion of the details of these (that would be of use to specialists in these areas) would need to be the subject of a separate paper. Both are highly complex science areas in their own right, and their treatments in the models are highly diverse.

- *(3.17) PSCs*

- *For the non-expert, elaborate on what is calculated assuming 'thermodynamic equilibrium'. Can mention formation of NAT/ice PSCs, and how these differ between models.*

  We have added a sentence explaining "thermodynamic equilibrium", and a further sentence on the PSCs.

- *It would make sense for this section to be near Sect. 3.11 (heterogeneous chemistry)*

  We have re-ordered the subsections.

- *(3.22) Ocean coupling*

- *This section should be combined with Sect. 3.12 (ocean surface forcing) or appear close to it.*

  We have combined the two subsections.

- *Mention also the sea ice modules / boundary conditions here.*

  We have added a sentence on sea ice.

- *(3.23) Solar forcing*

- *In cases where SW radiation and photolysis are not handled consistently, what are the radiation schemes for photolysis? For these photolysis schemes, can we assume the effects of the 11-yr solar cycle are not included?*

  A diversity of photolysis schemes are used, as is detailed in table S22. Some of them consider the solar cycle, others don't. We have rephrased this for clarity. The fact that shortwave radiation and photolysis have not been made consistent in several models does not imply there are significant inconsistencies in these models. It only means the schemes were developed independently. In response

to this question, several models have been re-classified as treating shortwave radiation and photolysis consistently.

- *(4) CCMI simulations*

- *P10 L15: 'Forcings are discussed briefly...' → 'The specific forcings imposed are discussed briefly...'.*

  Done.

- *P10 L18-20: This sentence is very unclear and should probably be separated into at least 3 sentences. Do you mean that ODS concentrations (or EESC), rather than emissions, peak around yr 2000? Which 'industrial emissions' do you mean? Separate the discussion of GHGs from ODSs and the other 'emissions' that are referred to. Should this sentence be in the Forcings section?*

  We agree that this sentence is displaced and "loose". We have cut it out. A more exhaustive discussion of emissions is in the "Forcings" section.

- *P10 L22: clarify that SD stands for "specified dynamics".*

  Done.

- *P10 L22: differences in dynamics between nudged and free-running models are not necessarily due to inherent dynamical biases in the model; they could also be due to the greater presence of internal variability in the free-running case.*

  Indeed. The opposite (reduced variability) can also be the case. We have rephrased this sentence.

- *P11 L3: which emissions?*

  We have clarified the statement.

- *P11 L5: I'm confused as to the exact forcings imposed here: GHGs? SSTs and sea ice for models not coupled to an ocean? ODSs (including those that are not GHGs)? NOx? NMHCs?*

  In this set of simulations, all assumptions about future climate forcings are changed from RCP 6.0/WMO (2011) to a different RCP (keeping WMO (2011) for ODSs). This includes changing ocean conditions accordingly, for those models that require that. We have added two sentences to this subsection.

- *P11 L9: 'sea surface' → 'sea surface and sea ice'.*

  Done.

- *P11 L11: only surface emissions, or also 3D emissions (e.g. aircraft NOx)?*

  Both. We have changed the sentence accordingly. This does not include lightning NOx emissions which are usually handled interactively though.

- *P11 L15: clarify that SSI stands for Spectral Solar Irradiance.*

  We have expanded the acronym.

- *(5) Forcings*

- *P12 L1-2: clarify that N2O and CH4 boundary conditions refer to surface mass mixing ratios.*

  Done. Actually the plots show volume mixing ratios.

- *P13 L9: add reference.*

  Done (WMO, 2015).

- *P13 L14: 'cause NOx emissions to peak and then decline' - clarify that this is only an assumption for the future.*

  We have rephrased this sentence to make this clear.

- *P13 L26: reference for MEGAN model*

  We have added a web link for MEGAN.

- *P13 L19-30: which dataset(s) is/are used for historic emissions in which biogenic emissions are not interactively computed?*

  The recommendation was to use biogenic emissions datasets of the modellers' choice, which was "preferably consistent with the model's meteorology". In reality, a range of different emissions was used.

- *P14 L10: reference appropriate table*

  Done.

- *P15 L1-15: much of this information is provided in Sect. 3.21. Instead of repetition, talk here about the time series of aerosol precursor emissions (e.g. the projected reduction in future aerosol emissions over certain regions as with ozone precursors).*

  We now include a mention of this effect. We now have a plot showing emissions of NOx and SO2, at different times throughout the REF-C2 period.

- *Tables*

- *Table 2: Is it possible to list (alongside the model names) the versions used for CCMVal-2, and, where relevant, the name of the ESM? Right now, there is little consistency.*

  In as far as Morgenstern et al. (2010) have given versions, we now state these versions in the table.

- *Table 4: caption: state that the numbers in the table stand for (I'm guessing) the number of ensemble members.*

  Done.

- *Table 5 and 6: why are some numbers in bold and in brackets? What does L39 stands for?*

  Numbers in brackets mean that these simulations are in progress (i.e. unavailable at the time of writing). "L39" stands for the 39-level version of the LMDz model. We have clarified these issues in the caption. The boldface will disappear for the final version.

- *Table S2 caption: clarify that the numbers represent number of grid boxes.*

  Done.

- *Table S5 caption: expand on abbreviations used in the table e.g. SL = semi-Lagrangian etc.*

  Done.

- *Table S5: for CESM models, CAM4 describes the atmospheric component but not the transport scheme (is it SL? please check.)*

  It's finite-volume (Neale et al., 2013). The entries have been changed accordingly.

- *Table S26: would make sense to keep this next to Table S5.*

  We have re-ordered the table in line with the reviewer's earlier comment.

- *All tables: please try to find more references for each model (e.g. for the aerosol schemes in Table S4).*

  We have added more references in several tables.

- *Appendices*

- *P22 L22-24: The two first sentences are already mentioned in the previous paragraph.*

  We have shortened this paragraph to remove the duplication.

- *P23 L15: citation for Wesely scheme.*

  Added. This scheme is also used by UMUKCA-UCAM and HadGEM3-ES; citations have been added.

- *P25 L29: why are two of the simulations in bold?*

  We have changed this to regular font.

- *P26 L24: elaborate.*

  We now state a reference (Jöckel et al., 2016) which has the details on this. Diagnostic tracers are the subject of a separate publication; we have therefore avoided discussing them here.

- *Technical comments*

- *P4 L12: misspelled 'oscillation'*

  Corrected.

- *P5 L18: 'included' → 'include'*

  That sentence has been rephrased.

- *P5 L27: 'Unified Model' → 'MetUM' for consistency*

  We have applied this name throughout the paper.

- *P9 L2: Remove comma after 'table S17'*

  Done.

- *P10 L8: 'increasing' → increasingly'*

  Done.

- *P19 L13: "Earth'" → "Earth's"*

  Done.

- *P24 L18: 'CTM be' → 'CTM can be'*

  Done.

- *Table S3 caption: 'CCM name' → 'model name'*

  Done.

---

## Author Response (AR2)

Dear Astrid Kerkweg,

Thanks a lot for accepting the paper subject to technical corrections. The uploaded manuscript addresses all but one of your remaining comments, listed below.

Page 7, l. 29: add closing bracket after "represented"

Done

Page 10, l. 15: delete brackets

This line does not contain brackets.

p. 15, l. 30/31: "These data sets are intended to be applied ...." ; Does this mean, that in fact they are not?

We have rephrased this to "These data sets are recommended to be applied…" Most models do not apply solar particle forcing, as is discussed elsewhere.

p. 16, l. 24: " ...to inform other papers": hopefully you want to inform scientists (or users of the data) and not papers ?

We have rephrased this to "inform authors of other papers".

Code availability: Maybe cite Table 1 here for the contact addresses?
Done.

References: it is not clear which of the Eyring et al., 2013 papers is 2013a and which is 2013b

We have added such identifiers to the year of publication (2013a and 2013b).

Please introduce the abbreviations for figure (Fig.), section (Sect.) etc, according to the author guidelines for manuscript preparation (http://www.geoscientific-model-development.net/for_authors/manuscript_preparation.html)

We now spell out all uses of "figure", "table", or "section".

Best regards,

Olaf Morgenstern.